

# The three-dimensional structure of fronts in mid-latitude weather systems as represented by numerical weather prediction models

Andreas A. Beckert[1], Lea Eisenstein[2], Annika Oertel[2], Tim Hewson[3], George C. Craig[4], and Marc Rautenhaus[1]

5  [1] Visual Data Analysis Group, Regional Computing Centre, Universität Hamburg, Hamburg, 20146, Germany
[2] Institute of Meteorology and Climate Research, Karlsruhe Institute of Technology, 76131, Karlsruhe, Germany
[3] European Centre for Medium-Range Weather Forecasts, Reading, RG2 9AX, United Kingdom
[4] Meteorological Institute, Ludwig-Maximilians-University Munich, Munich, 80333, Germany

10  *Correspondence to*: Andreas A. Beckert (andreas.beckert@uni-hamburg.de)



**Abstract.** Atmospheric fronts are a widely used conceptual model in meteorology, most encountered as two-dimensional (2-D) front lines on surface analysis charts. The three-dimensional (3-D) dynamical structure of fronts has been studied in the literature by means of "standard" 2-D maps and cross-sections and is commonly sketched in 3-D illustrations of idealized weather systems in atmospheric science textbooks. However, only recently the feasibility of objective detection and visual analysis of 3-D frontal structures and their dynamics within numerical weather prediction (NWP) data has been proposed, and such approaches are not yet widely known in the atmospheric science community. In this article, we investigate the benefit of objective 3-D front detection for case studies of extratropical cyclones and for comparison of frontal structures between different NWP models. We build on a recent gradient-based detection approach, combined with modern 3-D interactive visual analysis techniques, and adapt it to handle data from state-of-the-art NWP models including those run at convection-permitting kilometer-scale resolution. The parameters of the detection method (including data smoothing and threshold parameters) are evaluated to yield physically meaningful structures. We illustrate the benefit of the method by presenting two case studies of frontal dynamics within mid-latitude cyclones. Examples include joint interactive visual analysis of 3-D fronts and warm conveyor belt (WCB) trajectories, and identification of the 3-D frontal structures characterising the different stages of a Shapiro-Keyser cyclogenesis event. The 3-D frontal structures show agreement with 2-D fronts from surface analysis charts and augment the surface charts by providing additional pertinent information in the vertical dimension. A second application illustrates the effect of convection on 3-D cold front structure by comparing data from simulations with parameterised and explicit convection and shows that convection could strengthen the cold front. Finally, we consider "secondary fronts" that commonly appear in UK Met Office surface analysis charts. Examination of a case study shows that for this event the secondary front is not a temperature-based but purely a humidity-based feature. We argue that the presented approach has great potential to be beneficial for more complex studies of atmospheric dynamics and for operational weather forecasting.





## 1 Introduction

The concept of atmospheric fronts, first introduced by Bjerknes (1919), plays a prominent role in meteorology. They are
thought of as an interface separating two air masses of different density, mostly caused by temperature differences (Front - Glossary of Meteorology, 2022). Fronts are imaginary surfaces in three-dimensional (3-D) space, however, most commonly they are encountered as two-dimensional (2-D) lines on surface analysis charts, where they still frequently originate from manual analysis of different atmospheric variables. Despite the prevalence of 2-D surface fronts in meteorological practice, several studies have highlighted the impact of the vertical structure of fronts on surface weather (Bader et al., 1996; Browning
and Monk, 1982; Locatelli et al., 1994, 2005; Aemisegger et al., 2015). Hence, analysis of the full 3-D temporal evolution of frontal surfaces has great potential to be beneficial both for weather forecasting and research on atmospheric dynamics.

Here, we consider analysis of frontal dynamics for investigations including case studies and comparison of frontal structures between simulations from different numerical models. Analysis of 3-D frontal structures for such applications requires 3-D visualization and some objective feature detection method due to the difficulty of manual 3-D analysis on the one hand, and
the requirement of feature consistency across time and/or different data sets on the other. Such analysis and the benefits for weather forecasting and research gained from it have, to the best of our knowledge, not been thoroughly addressed in the literature. To fill this gap is the purpose of the present study.

Algorithms for 2-D objective front detection have been developed since the 1960's (e.g., Renard and Clarke, 1965; Huber-Pock and Kress, 1989; Jenkner et al., 2009). A widely cited method based on the third derivative of a thermal variable was
introduced by Hewson (1998), and recently extended from 2-D to 3-D by Kern et al. (2019). Kern et al. (2019) integrated the objective detection algorithm into the open-source meteorological interactive 3-D visualization framework "Met.3D" (Rautenhaus et al., 2015a, b; Met.3D – Homepage, 2022; Met.3D – Documentation, 2022) and demonstrated the feasibility of interactive 3-D visualization of frontal surfaces detected in numerical weather prediction (NWP) data from the European Centre for Medium-Range Weather Forecasts (ECMWF). In the present study, our objective is to address open issues about
the applicability of the method, and to demonstrate and evaluate its use for analysis of atmospheric dynamics and for examining other NWP datasets of different spatial resolution.

The methods based on Hewson (1998) and Kern et al. (2019) (as well as further detection methods proposed in the literature) build on extracting frontal feature candidates from fields of the third derivative of a thermal variable (cf. Thomas and Schultz, 2019a) that typically are smoothed to some extent to remove high-frequency fluctuations. The feature candidates are then
filtered according to some filter criteria (most prominently, a so-called "thermal front parameter", TFP, and the frontal strength) to yield the final frontal features. Two challenges arise when applying such an approach to modern NWP data. First, the current trend towards convection-permitting kilometer-scale resolution in NWP models leads to more small-scale fluctuations in the gradient fields. The question arises whether the existing approaches still extract meaningful structures that represent a frontal surface. A related issue is that smaller numerical differences between the values of neighboring grid cells (caused by smaller
grid-point spacing) require care to avoid numerical artefacts when computing higher-order derivatives (cf. Jenkner et al., 2009).



Second, threshold values for filtering of feature candidates need to be selected carefully to yield physically interpretable structures. In the literature addressing 2-D front detection, such thresholds have been set to "hard" thresholds, i.e., fixed values suitable for the data and elevation level used. Such thresholds may not be generalized across different model resolutions and vertical elevations (Hewson, 1998). Furthermore, hard thresholds can lead to undesired "holes" in the resulting frontal surfaces,

e.g., where frontal strength or TFP are only slightly below the chosen threshold. Therefore, Kern et al. (2019) proposed a fuzzy filtering method with upper and lower filter thresholds, between which the frontal features are gradually faded. However, past literature focused little on the filtering process and how to select suitable thresholds.

For analysis of the detected 3-D features, recent advances in 3-D computer graphics and visualization bear large potential for intuitive, rapid interpretation in the context of the underlying atmospheric situation. Such techniques are not yet widely used

in weather forecasting and research, with reasons including a lack of suitable software tools and a lack of literature demonstrating the benefit of 3-D visual analysis (Rautenhaus et al., 2018). An overview of the current state of the art in visualization in meteorology has recently been provided by Rautenhaus et al. (2018); recent examples of 3-D visual analysis being applied to meteorological research include the studies by Rautenhaus et al. (2015b), Orf et al. (2017), Kern et al. (2018, 2019), Bader et al. (2020), Meyer et al. (2021), Bösiger et al. (2022), and Fischer et al. (2022).

In the present study, we further contribute to the literature on benefits of atmospheric feature detection and 3-D visual analysis for weather forecasting and research and address the following objectives:

(a) Advance the Kern et al. (2019) approach to objectively detect 2-D and 3-D frontal structures independently of the grid point spacing of the input NWP data, to be able to compare frontal structures between, for instance, different model resolutions (e.g., in convection-permitting vs. convection-parameterized simulations), different ensemble members or different cases. Our

goal is to shed light on the smoothing and filtering processes in the detection method and to study the sensitivity of changing smoothing parameters on the resulting detected fronts: Which smoothing parameters yield meaningful 3-D structures, and how do filtering thresholds need to be chosen accordingly?

(b) Evaluate the benefit of 3-D interactive visual analysis (IVA) of the detected frontal structures for the analysis of midlatitude cyclones. We focus on two case studies (cyclone *Vladiana*, crossing the North Atlantic in September 2016, and winter storm

*Friederike*, hitting Germany in January 2018) and address the following questions: Can we confirm known knowledge about the 3-D dynamical structure of fronts and related warm conveyor belts (WCB) by means of 3-D IVA? How can the characteristic frontal development stages of a Shapiro-Keyser cyclone be distinguished in 3-D? How do 3-D frontal structures differ in (higher resolution) convection-permitting vs. (lower resolution) convection-parameterizing simulations? How do the detected 3-D structures compare to official analyses by the UK Met Office, in particular with respect to "secondary warm

fronts" often observed in UK Met Office charts?

In this study, we build upon the Kern et al. (2019) approach integrated into Met.3D (Met.3D – Code Repository, 2022). This facilitates straightforward use of the existing interactive 3-D visualization techniques in the software, including a "bridge from 2-D to 3-D" (cf. Rautenhaus et al., 2015a) to combine well-proven 2-D views with new 3-D perspectives. Our method is flexible with respect to the input data, for the presented case studies we use forecast and reanalysis data from ECMWF with a



horizontal grid spacing between 0.15˚ and 0.25˚, and data from the limited-area model COSMO (Consortium for Small-scale Modeling; Baldauf et al., 2011; Doms and Baldauf, 2018) with a horizontal grid spacing of 0.02˚.

The article is structured as follows. Section 2 introduces the underlying objective front detection approach by Hewson (1998), its extension to 3-D by Kern et al. (2019), and our enhancements for detection of fronts in kilometer-scale resolution data. Section 3 introduces the case studies and the data used for their visualization. In Section 4, we discuss which thermal variable

is suitable for the approach and how sensitive detected fronts are with respect to different data resolutions and smoothing parameters. Section 5 presents the case studies to evaluate the benefit of 3-D front analysis for weather forecasting and research, and Section 6 summarizes and concludes the study.





## 2 Method and implementation

Our algorithm follows the 2-D detection algorithm originally introduced by Hewson (1998) and extended to 3-D by Kern et al. (2019). We briefly explain the basics of the algorithm and focus on the parts that have been adapted for this study. For further details we refer to Hewson (1998) and Kern et al. (2019). In the following, we describe and illustrate the conceptual and mathematical basis (Section 2.1), the required filtering process for frontal candidates (Section 2.2), and some implementation details we consider important (Section 2.3).

### 2.1 Conceptual and mathematical basis

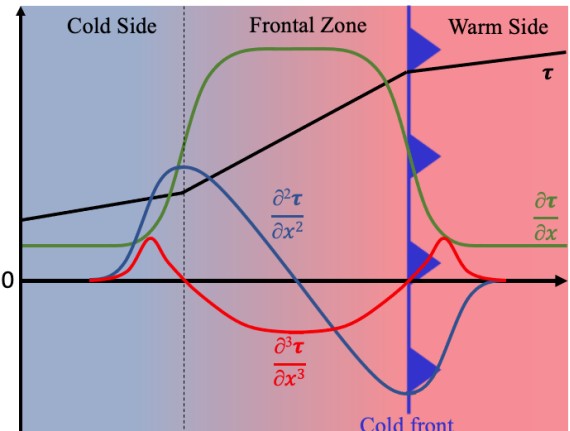

**Figure 1. Illustration of the thermal-gradient-based detection method, using a simplified straight front and following Hewson (1998) and Kern et al. (2019). The goal is to determine the warm air boundary of the frontal zone (i.e., the region of increased thermal gradient, cf. the green line). This boundary corresponds to the third derivative (red line) of a thermal variable $\tau$ (black line) being zero, under the condition that the second derivative of $\tau$ (blue line) is negative.**

Figure 1 illustrates the method. The goal is to detect the horizontal warm air "boundaries" of frontal zones, i.e., regions with a strong horizontal gradient of a thermal variable $\tau$ (black line). In the simplified 1-D example shown in Figure 1, the first partial derivative of $\tau$ with respect to the spatial dimension $x$ $(\partial\tau/\partial x)$ changes rapidly on both the warm and cold air boundaries of the frontal zone, with a maximum in between. Hence, the third derivative $\partial^3\tau/\partial x^3$ can be used to detect the locations of maximum gradient change; the locations where it is zero and the second derivative $\partial^2\tau/\partial x^2$ is negative coincide with the warm air boundary of the frontal zone (Hewson, 1998). In the general 2-D case, points on a frontal line need to fulfil the "front location equation" (cf. Hewson, 1998) to account for curved fronts and corresponding along-front thermal gradients:

$$L_\tau \equiv \frac{\partial(|\nabla_h|\nabla_h\tau||)_s}{\partial\hat{s}} = 0, \qquad (1)$$

with:

$$\hat{s} = \pm\frac{\nabla_h|\nabla_h\tau|}{|\nabla_h|\nabla_h\tau||}$$





Here, $\nabla_h$ denotes the horizontal derivative and $\hat{s}$ is a unit axis (which possesses an orientation but no direction) oriented along $\nabla_h|\nabla_h\tau|$. To derive 3-D frontal surfaces the approach is extended to 3-D as proposed by Kern et al. (2019). In short, the front location equation Eq. (1) is computed at every grid point of the gridded dataset; then "candidates" of frontal features are

obtained by computing 3-D isosurfaces of $L_\tau = 0$ using a contouring algorithm such as Marching Cubes (Lorensen and Cline, 1987). This results in a large number of potential frontal surfaces; to obtain meaningful structures the feature candidates need to be filtered according to additional diagnostics including the strength of the thermal gradient within the frontal zone. For details, we refer the reader to Kern et al. (2019, their Sect. 4). Note that only the horizontal gradient of the thermal variable is considered in this process, see Kern et al. (2019) for a discussion of the inclusion of vertical contributions.

## 2.2 Filtering

To obtain meaningful frontal surfaces (or frontal lines in the 2-D case), the feature candidates need to be filtered. Hewson (1998), following Renard and Clarke (1965), suggested to filter according to the thermal front parameter TFP, as well as to a frontal strength value estimated by the local thermal gradient at the frontal feature. The latter was improved by Kern et al. (2019) to estimate frontal strength by computing an average thermal gradient along "normal curves" traced through the frontal

zone (basically streamlines computed on the gradient vector field). Here, we generalize these two filters to more generic types of filter mechanisms that can be interactively modified and combined during the analysis to investigate different aspects of the data:

a) Masking: The feature candidates are filtered according to an arbitrary 3-D scalar field that is sampled (i.e., interpolated) at all feature locations (e.g., if isosurfaces are extracted using Marching Cubes, at all vertices of the 
isosurface). User-defined thresholds of the scalar field are used to keep or discard features.

b) Frontal zone traversal: the frontal zone is traversed along "normal curves" started at feature candidate vertices and computed on the thermal gradient field (Kern et al., 2019); an arbitrary 3-D scalar field is sampled along the normal curves and filtering thresholds are based on the obtained samples.

The generalization allows us, in addition to filtering with respect to TFP and frontal strength, to add filters that facilitate focus
on the contribution of further quantities, including, for example, humidity and elevation. This way, we can eliminate, for example, pure "humidity fronts" by tracing the changes in (dry) potential temperature ($\theta$) along the normal curves. TFP and frontal strength, however, remain to be the core filters.

### 2.2.1 TFP Masking

TFP is a masking filter. Note that computing isosurfaces of $L_\tau = 0$ results in front feature candidates at both cold and warm
side of the frontal zone. Since we are interested in the warm side only (cf. Renard and Clarke, 1965), cold side feature candidates need to be discarded. We follow the approach of Hewson (1998) and use the TFP filter, first introduced by Renard and Clarke (1965). The TFP filter is defined as:



$$TFP_\tau \equiv -\nabla_h |\nabla_h \tau| \cdot \frac{\nabla_h \tau}{|\nabla_h \tau|} > K_1, \tag{2}$$

where $K_1$ is a used-defined threshold. This equation can also be interpreted as the "negative curvature" of the thermal front
parameter field (Kern et al., 2019), being positive at the warm side of the frontal zone and negative at the cold side. To obtain
only frontal feature candidates at the warm side of the frontal zone, $K_1$ must be at least zero. Hewson (1998) suggested a
slightly positive value for $K_1$ to eliminate spurious frontal pieces.

### 2.2.2 Frontal Strength

The normal-curve-based frontal strength filter is applied to the remaining warm air side frontal candidates. We follow Kern et
al. (2019) and estimate the frontal strength of the filter variable as "the average thermal gradient along a curved path through
the frontal zone from the warm to the cold-air side". The frontal strength filter $S_\tau$ is defined as:

$$S_{\tau|frontal\ zone} \equiv \int_{NC} |\nabla_h \tau| \, ds > K_2, \tag{3}$$

The integration through the frontal zone starts at the warm side of the frontal zone and stops once a "normal curve" reaches
the cold side of the frontal zone (where $L_\tau$ again is zero). The threshold $K_2$ is used to eliminate weak fronts below a user-
defined frontal strength.

### 2.2.3 Fuzzy filtering

Usage of distinct threshold values for $K_1$ and $K_2$ results in "hard" boundaries of the generated features. Such visualization can
be misleading since a viewer can interpret distinct feature boundaries into the depiction (including, e.g., "holes" in the front
surfaces where, e.g., frontal strength is just below the chosen threshold). For fronts, however, this is not the case, as thermal
gradients are gradually decreasing in space. Kern et al. (2019) suggested a "soft" (or "fuzzy") filtering by providing two
thresholds for each filter, between which opacity is faded from zero (completely transparent) to one (completely opaque). The
feature candidates are subsequently rendered using the obtained opacity, resulting in "fuzzy" edges that visually indicate, e.g.,
a decreasing thermal gradient. The approach can also facilitate a visual distinction between weak fronts and strong fronts.
When multiple filters are used in our implementation, every filter has individual threshold interval settings, and opacity
information are accumulated accordingly.

### 2.3 Supported data and methodological details

The presented algorithm supports gridded data on horizontally regular and rotated latitude-longitude grids. In the vertical, the
implementation can handle both pressure levels and model levels. For this study, we use data from the operational ECMWF
high-resolution forecast (HRES) with 137 vertical model levels, horizontally interpolated to a regular grid with a grid point
spacing of 0.15° in both latitude and longitude, data from the global reanalysis ERA5 (Hersbach et al., 2020) (also 137 vertical
model levels, interpolated to a horizontal grid spacing of 0.25°), and data from the COSMO model (Consortium for Small-
scale Modeling; Baldauf et al., 2011; Doms and Baldauf, 2018), available on a rotated latitude-longitude grid with 60 vertical



model levels and a horizontal grid point spacing of 0.02° in both dimensions. The algorithm has been integrated into the interactive visualization framework Met.3D (Rautenhaus et al., 2015a) and is being made available as open-source.

In the following, we describe methodological details we deem important for understanding our approach. Figure 2 illustrates the main steps of the front detection process. For simplicity, the process is described for 2-D frontal lines:

1. Choice of a thermal input field $\tau$ (e.g., wet-bulb potential temperature).

2. Smoothing of $\tau$ (and further input fields used for filtering) to a user-define length scale.

3. Computation of horizontal gradients $\nabla_h \tau$.

4. Computation of the magnitude of horizontal gradients $|\nabla_h \tau|$.

5. Computation of the horizontal gradient of the magnitude of horizontal gradients $\nabla_h |\nabla_h \tau|$.

6. Evaluation of the front location equation Eq. (1) and computation of the zero isolines to obtain feature candidates.

7. Application of the TFP masking filter.

8. Application of frontal strength and further "normal curves" filters.

In the 2-D example in Figure 2, the 850 hPa pressure level is used. One important design decision for the 3-D variant of the algorithm is the choice of the vertical coordinate, as the numerical computations need to be implemented accordingly. For this study, we consistently use pressure as the vertical coordinate, i.e., all horizontal computations are evaluated on levels of constant pressure. This is also consistent with Met.3D's use of pressure as vertical coordinate.

### 2.3.1 Smoothing

NWP data, especially at kilometer-scale resolution, include convective and thermal processes that are much smaller in scale than atmospheric fronts (Keyser and Shapiro, 1986). To obtain frontal features that meaningfully represent a scale of interest (e.g., synoptic-scale fronts), it is advisable to smooth small scale thermal fluctuations in the thermal input field. Previous studies have used simple smoothing filters like a weighted moving average of neighboring grid points (e.g., Jenkner et al., 2009), well known from image processing (Davies, 2017). Kern et al. (2019) point out that for data on a regular longitude-

latitude grid, however, geometric distance between grid points varies with latitude, requiring usage of a smoothing filter that considers all grid points based on a specified geometric smoothing distance. They propose usage of a 2-D Gaussian smoothing kernel.

In our implementation, the smoothing distance is a user-defined method parameter that can be interactively changed in the analysis process. A disadvantage of a Gaussian smoothing filter, however, is its computational complexity that increases

quadratically with smoothing distance – an important aspect for interactive use. We hence also provide an approximative smoothing method, the "fast almost-Gaussian filtering" presented by Kovesi (2010). The method uses a specified number of averaging passes. More averaging passes increase the accuracy of the approximative algorithm compared to Gaussian smoothing, but at the cost of increasing computation time and, also an important aspect, with an increased number of averaging passes the effect of "smoothing over the data field edges" propagates further into the data field center (Kovesi, 2010). In our

implementation, the smoothing computation complexity depends linearly on the averaging passes and the smoothing distance.





We find that three averaging passes are a reasonable tradeoff between accuracy, computation time and keeping the edged effect small. For illustration, we measured the performance of both smoothing algorithms on six cores of an AMD Epyc 7542 32-core processor with 2.9 GHz. In this setup, it takes about 29.5 seconds to apply a horizontal Gaussian smoothing with a smoothing distance of 100 km to a 3-D data field of 1800 x 1800 horizontal grid points with a horizontal grid spacing of 0.02°
and 31 vertical level. For the same data field, the approximative algorithm requires 3.9 seconds. Both algorithms are optimized for OpenMP and run in parallel.

### 2.3.2 Numerical implementation

For the computation of horizontal gradients, we use first order finite central differences and at boundaries first order finite right and left differences, respectively. As described above, we use pressure as the vertical coordinate and hence need to adapt
the computations for data available on hybrid sigma pressure model levels or geometric altitude model levels. This leads to an additional coordinate transformation term (cf. Etling, 2008, p.129–131) in the derivatives. The horizontal gradient in pressure coordinates $|_p$ of the thermal variable $\tau$ is obtained from the partial derivative in longitudinal direction on the original coordinate system $|_\sigma$ and an additional transformation term. The gradient component in the longitudinal direction hence becomes:

$$\frac{\partial \tau}{\partial lon}\Big|_p = \frac{\partial \tau}{\partial lon}\Big|_\sigma + \frac{\partial \tau}{\partial p}\Big| \cdot \frac{\partial p}{\partial lon}\Big|_\sigma \tag{4}$$

And the latitudinal component:

$$\frac{\partial \tau}{\partial lat}\Big|_p = \frac{\partial \tau}{\partial lat}\Big|_\sigma + \frac{\partial \tau}{\partial p}\Big| \cdot \frac{\partial p}{\partial lat}\Big|_\sigma \tag{5}$$

Care needs to be taken for numerical implementation of equations 1-5. For numerical stability reasons, Hewson (1998) computed $\hat{s}$ as a "five-point-mean axis" – an average orientation axis derived from the gradient at the corresponding grid point
and at the four surrounding grid points (for details cf. Hewson, 1998). We encountered challenges with this approach:

    a)    The studies by Hewson (1998) and Kern et al. (2019) used gridded data with a regular horizontal grid point spacing on the order of 50 km (0.5°) to 100 km (1°). At the time of writing, current (e.g., limited area) NWP models use finer grid spacings, e.g., the regional forecast model of the German Weather Service (DWD) runs with a horizontal grid spacing of 0.02°. At such resolutions and depending on the smoothing distance of previously applied smoothing, the
250        differences between data values at neighboring grid cells tend to be very small – in such cases, no numerically stable orientation of the five-point mean axis can be obtained.

    b)    Analogous to the above reasons for use of a distance-based Gaussian smoothing filter, the dependence of geometric distance between neighboring grid points on latitude leads to inconsistent calculations of the five-point-mean axis.

    c)    The distance between neighboring grid cells depends on the grid-point spacing of the specific dataset used. To
255        compare fronts in different model simulations with a different grid-point spacing it is inconvenient to use a grid-point based approach, because the distance of the neighboring grid cell changes with changing model resolutions.



Instead of taking the neighboring grid points to calculate the five-point-mean axis, we propose to use interpolated values at a specified distance to the considered central grid point. This improves numerical stability, makes the computation independent of geographic location, and facilitates objective comparison of frontal features obtained from NWP datasets with different

grid-point spacings. From our experiments, we find that using a distance for the five-point-mean axis computation of half of the smoothing distance works well.





**Figure 2. Step-by-step illustration of the 2-D front detection method. In the example, objective fronts are based on the 850 hPa wet-**
**bulb potential temperature field ($\theta_w$) from the ECMWF HRES forecast (horizontally regular grid point spacing of 0.15° in both**
**longitude and latitudes) from 18 January 2018, 12:00 UTC. Fronts are "fuzzy filtered" using a fade-out range for TFP of 0.2 – 0.4**
**K (100km)$^{-2}$ and for frontal strength of 0.6 – 1 K (100 km)$^{-1}$.**



## 3 Case studies and applied data

We illustrate the capabilities and value of 3-D front detection and visualization using data from simulations of two extratropical
cyclones. The first case, cyclone *Vladiana*, occurred in the North Atlantic in September 2016. The second case, winter storm
*Friederike*, took place in Western Europe in January 2018. This section briefly describes the synoptic situation of the respective
cyclones and introduces the used datasets. We also briefly revisit the underlying meteorological theory.

### 3.1 Meteorological theory

The frontal structure of extratropical cyclones is a key feature for the analysis of their development. Typically, extratropical
cyclones are classified as either classical Norwegian cyclones (Bjerknes, 1919) or (the later proposed) Shapiro-Keyser cyclones
(Shapiro and Keyser, 1990). The development of both cyclone types is classified into four characteristic stages. A cyclone first
develops along a frontal wave as a small disturbance near the surface (stage I in both models). Meanwhile, this disturbance
strengthens and extends to higher elevations, the cyclone starts to rotate cyclonically and forms a warm sector (stage II). In
stage II the warm sector has its maximum size and maximum energy conversion. For Norwegian cyclones the displacement
speed of the cold front is faster than of the warm front, the warm sector diminishes (stage III). The fronts occlude forcing the
air to rise before the cyclone finally dissipates (stage IV). In contrast, a Shapiro-Keyser cyclone develops a frontal fracture in
stage II separating the cold front from the warm front. While the cold front is usually weaker than in Norwegian cyclones
(Schultz et al., 1998), the warm front is north of the cyclone center and starts wrapping around it bending backwards, hence
also called bent-back front (stage III). This stage is also called "T-bone structure". With the warm front wrapping around the
cyclone center, a warm seclusion occurs (stage IV) before the cyclone decays. More recent literature purpose an extension of
the four stages by three additional stages, the diminutive frontal wave stage and frontal wave stage which occur before stage I
and a decay stages after stage IV (Hewson and Titley, 2010). However, in this publication we focus on the initially proposed
four stages of the Shapiro-Keyser cyclone model.

Both cyclone models can be accompanied by coherent circulation features called conveyor belts. The cold conveyor belt occurs
ahead of the warm and occlusion front, usually remaining below 850 hPa. It is often associated with high wind speeds in later
stages, typically south-west of the cyclone center. The warm conveyor belt (WCD, cf. Eckhardt et al., 2004; Madonna et al.,
2014) occurs ahead of the cold front near the surface in early stages and is also associated with high wind speeds. It typically
ascends at least 600 hPa in the warm sector and over the warm front and often splits into anticyclonically and cyclonically
turning branches (Martínez-Alvarado et al., 2014).

### 3.2 *Vladiana*

The extratropical cyclone *Vladiana* occurred during the North Atlantic Waveguide and Downstream Impact Experiment
(NAWDEX, Schäfler et al., 2018). *Vladiana* formed on 22 September 2016 near Newfoundland and the frontal wave
intensified while moving eastwards across the North Atlantic. As the cyclone continued to move north-eastward, it





strengthened until it reached its pressure minimum of 975 hPa at 18:00 UTC on 23 September. On 24 September the cyclone

reached Iceland and became stationary. Figure 3 shows a horizontal section of $\theta_w$ with detected 2-D fronts at 850 hPa, as well as 3-D fronts on 23 September 2016, 06:00 UTC5.3. The frontal analysis of this case study builds upon previous studies of *Vladiana* and its associated WCB ascent (Kern et al., 2019; Oertel et al., 2019, 2020; Choudhary and Voigt, 2022). Based on the results of Oertel et al. (2019), we evaluate the conceptual model of 3-D fronts and WCB ascent (Section 5.1). In addition, we compare the cold frontal structure of simulations with explicit and parameterized convection (Section 5.3).

For our analysis we use ECMWF HRES analysis data with parameterized convection, a convection-permitting simulation with the limited-area model COSMO, and UK Met Office surface analysis charts. Initial and lateral boundary conditions of the COSMO simulation were taken from the ECMWF HRES analysis (see Oertel et al., 2019, 2020 for a detailed description of the simulation set-up). The COSMO simulation includes online trajectories (cf. Miltenberger et al., 2013) which were used to selected strongly ascending trajectories with ascent rates of at least 600 hPa in 48 h, here referred to as WCB trajectories

(Oertel et al., 2019, 2020). For the evaluation of the conceptual model of 3-D fronts and WCBs, WCB trajectories that ascend at least 25 hPa in 2 h at 06:00 UTC, 23 September 2016 were selected.

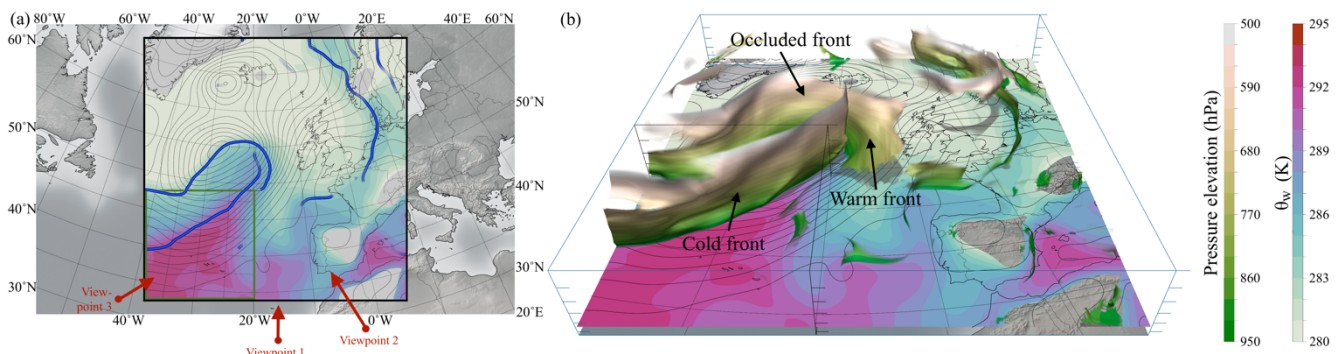

**Figure 3. Cyclone *Vladiana* on 23 September 2016, 06:00 UTC. (a) Detected 2-D fronts at 850 hPa (blue lines), $\theta_w$ at 950 hPa (colours, in K), and mean sea level pressure (black contour lines, every 2 hPa) from the COSMO simulation (black frame shows domain**

**boundaries; green frame shows the selected subregion for studying convection in the vicinity of the cold front (cf. Sect. 5.3))5.3. Red arrows show different viewpoints used in Figure 9. (b) Detected 3-D fronts between the surface and 500 hPa, augmented by a horizonal map showing $\theta_w$ at 950 hPa and mean sea level pressure (black contour lines, every 2 hPa).**

### 3.3   *Friederike*

The extratropical cyclone *Friederike* (called *David* in Great Britain) passed over western Europe from 17 to 18 January 2018.

The cyclone had formed east of Florida on 15 January 2018, then moved northwards along the coast of Newfoundland before it passed the North Atlantic Ocean and first hit Europe at the west coast of Ireland 17 January 2018. During its passage across the North Atlantic, the cyclone strengthened, and its core pressure dropped to 985 hPa. The cyclone moved from Ireland across northern England and the North Sea, reaching the north of the Netherlands on 18 January 2018, 09:00 UTC with a core pressure of 976 hPa. From there, the cyclone moved further east and passed northern Germany until it reached the border of Poland on

18 January 2018, 18:00 UTC and dissipated in the following days. As a result of the cyclone, high wind speeds were registered with gusts up to 203 km/h in the Harz Mountains, 144 km/h at the North Sea coast of the Netherlands, and 138 km/h in lowlands





of the Netherlands and central part of Germany (Wandel et al., 2018). Surface analysis charts of the UK Met Office (not shown here) indicate that this is a Shapiro Keyser Cyclone (Shapiro and Keyser, 1990). Our 2-D front algorithm detects some of the characteristic frontal features of the Shapiro-Keyser cyclone, including the frontal wave stage, frontal fracture, and t-bone

structure (Figure 4). This case will allow us for the first time (to our knowledge) to extract and visualize the 3-D frontal structure of a Shapiro-Keyser cyclone directly from NWP data and to evaluate the time evolution in comparison to the idealized model (Section 5.2). In Section 5.4 we analyse the occurrence of secondary warm frontal structures as often present in surface analysis charts of the UK Met Office.

Here we use the ERA-5 reanalysis and ECMWF HRES forecast data. ERA-5 reanalysis is used to visualize the temporal

development of 2-D (Figure 4) and 3-D (Figure 10) fronts. The ECMWF HRES forecast data was initialized on 18 January 2018, 00:00 UTC. For the analysis of secondary frontal structures, fronts extracted from the UK Met Office surface analysis charts supplement the ECMWF HRES forecast.

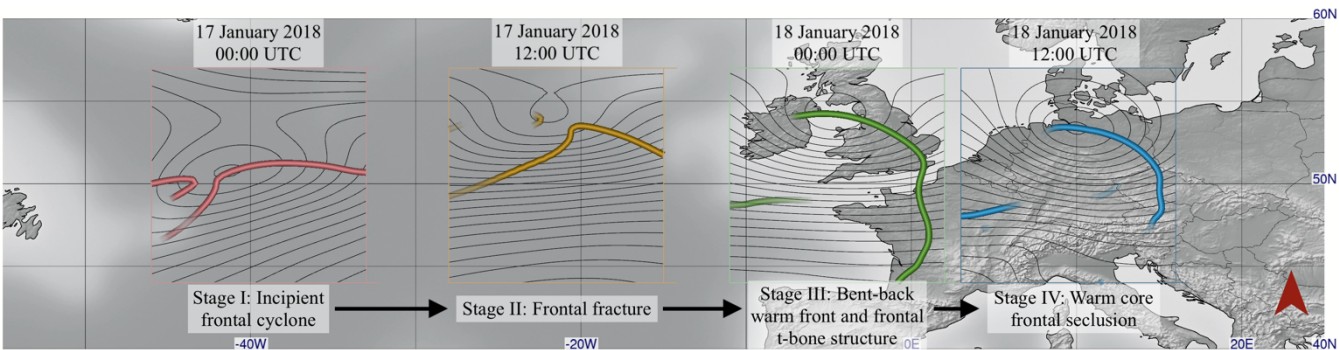

**Figure 4: Successive time steps of objective 2-D frontal structures showing the temporal development of *Friederike* (17 and 18**
**January 2018), as detected in ERA-5 reanalysis data at 750 hPa and surface pressure (black lines). The displayed time steps are approximately assigned to the four ideal development stages of the Shapiro-Keyser cyclone model (Shapiro and Keyser, 1990). We find that not all characteristics of the individual stages can be observed in 2-D. As shown in the following, 3-D front detection is required to observe all characteristics (cf. Figure 10).**





## 4 Thermal quantity, smoothing length scale, and filter parameters

To successfully apply front detection for case studies, three important aspects need to be considered: Which thermal quantity should be used for detection? Which smoothing distance should be applied to the data? How do filter thresholds need to be adjusted (also with respect to the smoothing distance)?

### 4.1 Choice of thermal quantity

We first discuss the role of the chosen thermal quantity. Three candidates have frequently been used in the literature: (dry)
potential temperature ($\theta$), wet-bulb potential temperature ($\theta_w$), and equivalent potential temperature ($\theta_e$). There is an ongoing discussion in the scientific community regarding which thermal quantity is best suited to detect fronts (e.g. Sanders and Doswell, 1995; Hewson, 1998; Berry et al., 2011; Schemm et al., 2018; Thomas and Schultz, 2019a, b). The following provides a brief overview of the potential thermal quantities and their advantages and disadvantages.

The dry potential temperature $\theta$, reflects the original, purely temperature-based, definition of fronts and is most convenient
from a rigorous dynamical point of view (Hewson, 1998). However, it is not conserved in moist processes, which often occur along fronts (Browning and Roberts, 1996). Alternative thermal quantities are $\theta_w$ or $\theta_e$, which are both conserved in the reversible diabatic processes of evaporation and condensation (Thomas and Schultz, 2019b). Since both quantities have a one-to-one relationship (each $\theta_w$ value matches a unique $\theta_e$ value and vice versa; Bindon (1940)), they share the same advantages and disadvantages for front detection (Thomas and Schultz, 2019b). In the following, we consider only $\theta_w$, but the arguments
should also be valid for $\theta_e$. The inclusion of humidity can help to better diagnose weak temperature gradients because humidity and temperature gradients are usually correlated, resulting in stronger $\theta_w$ gradients compared to $\theta$ gradients (Jenkner et al., 2009). However, if humidity and temperature are not correlated, gradients of $\theta_w$ could be weaker than gradients of $\theta$. This may result in $\theta_w$ fronts being weaker than $\theta$ fronts, up to not being detected at all. Furthermore, in regions with humidity gradients but without temperature gradients, purely humidity-based fronts can be detected. Therefore, Thomas and Schultz (2019)
recommended examining the temperature and moisture fields separately when analysing frontal structures. On the other hand, Berry et al. (2011) found that in their study $\theta_w$ provided the closest match to manually prepared front analysis. In our experience, $\theta_w$ is best suited to detect continuous fronts and closely matches the frontal analysis provided by UK Met Office (Figure 13). Note that some of the previously mentioned disadvantages of $\theta_w$ can be eliminated in our front algorithm. To facilitate the distinction between humidity- and temperature-based fronts, the implementation allows the mapping of different
quantities on frontal surfaces as well as filtering of fronts according to multiple variables. Mapping the total change of $\theta$ or specific humidity within the frontal zone could help to distinguish between humidity- and temperature-based fronts. If desired, fronts can be filtered according to $\theta$ or humidity gradients within the frontal zone, which can help to eliminate purely temperature or humidity based fronts (Hewson and Titley, 2010).





### 4.2 Recommendations for filter thresholds and sensitivity of fronts to different smoothing length scales

The number of detected frontal features depends on filter thresholds and the smoothing length scale applied to the input fields. Depending on the scale of interest for the analysis, the horizontal smoothing length scales is chosen. The question arises which filter thresholds for TFP and frontal strength filters should be recommended, and how these values depend on the smoothing length scale. In this section, we explore these method parameters and provide recommendations. We first investigate how smoothing length scale affects the magnitude and distribution of TFP values, then we consider magnitude and distribution of

frontal strength $|\nabla_h \theta_w|$. We present distributions of TFP and frontal strength obtained from 24 time steps of hourly ECMWF HRES forecast data on 18 January 2018 covering the area shown in Figure 2. The presented distributions provide guidance on the choice of suitable values for different smoothing length scales.

### 4.2.1 Dependence of filter thresholds $K_1$ and $K_2$ on smoothing length scale

Figure 5 shows the relative frequency of TFP values over the analysed area and for three different horizontal smoothing length

scales of 100 km, 50km, and 30 km. Large horizontal smoothing length scales result, in general, in lower TFP values and vice-versa. With large smoothing applied, strong horizontal gradients are weakened, resulting in smaller horizontal gradients. The magnitude of the horizontal gradients is inversely proportional to the length scale of the horizontal smoothing, and the filter thresholds need to be adjusted accordingly. Table 1 provides our recommendations for fuzzy TFP filter thresholds for the discussed smoothing scales.


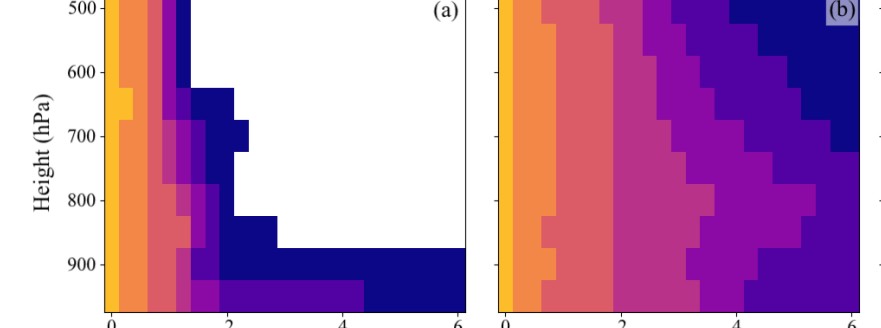

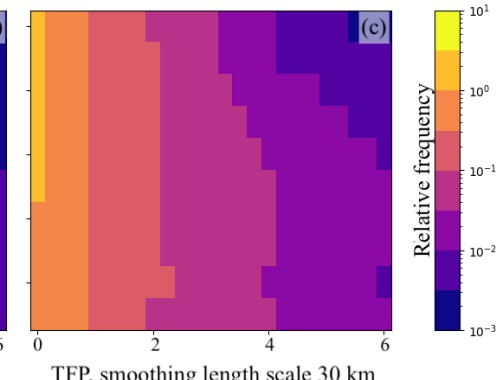

**Figure 5. Distribution of the thermal front parameter (TFP) between 950 - 500 hPa and for different smoothing length scales: (a) 100 km, (b) 50 km, and (c) 30 km. The distribution of TFP is relevant for selecting appropriate ranges to filter frontal candidates. Histograms show the relative frequency of occurrence of TFP values as computed from to hourly ECMWF HRES forecast data**

**(horizontally regular grid point spacing of 0.15° in both longitude and latitudes) from 18 January 2018, in the geographic region shown in Figure 2.**

Figure 6 shows the relative frequency of $|\nabla_h \theta_w|$ for same smoothing length scales as above, although this time only considering values at grid points within the frontal zone (i.e., where $L_\tau$ (Eq. 1) $> 0$). The same effect encountered for TFP can be observed, the horizontal smoothing length scale alters the relative frequency of $|\nabla_h \theta_w|$ as well. In general, $|\nabla_h \theta_w|$ decreases with





increasing horizontal smoothing length scale. As for TFP, it is necessary to adapt frontal strength filter thresholds to the chosen

horizontal smoothing length scale. Table 1 provides guidance.

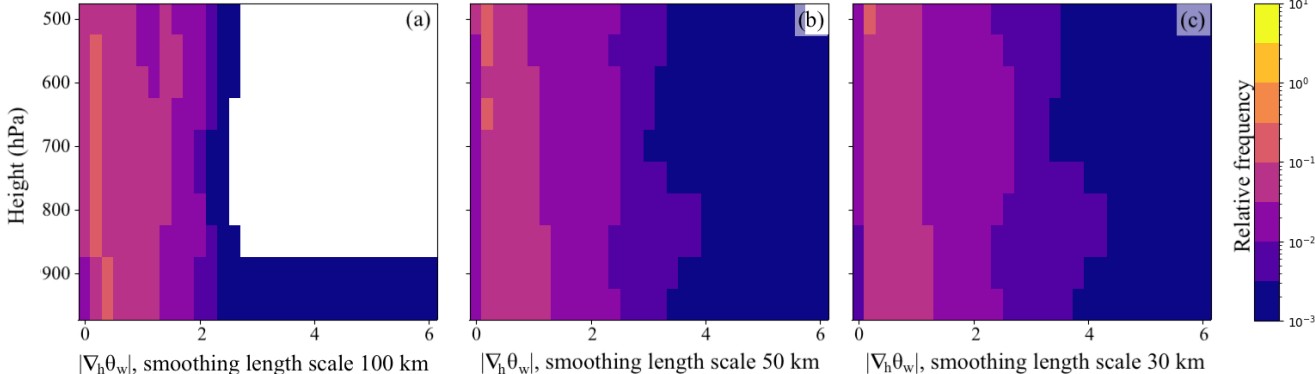

**Figure 6. Distribution of $|\nabla_h \theta_w|$ within frontal zones between 950 - 500 hPa and for different smoothing length scales: (a) 100 km, (b) 50 km, and (c) 30 km. The distribution of $|\nabla_h \theta_w|$ is relevant to select appropriate ranges for filtering frontal candidates. Same ECMWF HRES forecast data as in Figure 5.**

### 4.2.2 Example: Impact of filtering and smoothing on detected frontal features

As mentioned above, NWP data at kilometer-scale resolution includes convective and thermal processes that are much smaller in scale than atmospheric fronts (Keyser and Shapiro, 1986). If the focus of an analysis is on large scale frontal features, e.g., for large-scale weather analysis, the thermal variable can be smoothed with a distance between 50 km and 100 km. If smaller-

scale frontal surface phenomena, e.g., surface precipitation, are of interest, the smoothing distance can be reduced to a few kilometers. However, it should not be less than the grid spacing of the thermal input variable. In the following, we demonstrate how different smoothing length scales and filter thresholds impact the resulting frontal features. In particular, we show how different frontal strength filters can help distinguish between different front types (temperature- and humidity-based fronts). For this analysis we use ECMWF HRES forecast data for *Friederike* with a horizontal grid point spacing of 0.15°.

Figure 7a extends the 2-D visualization of Figure 2h to 3-D, depicting the full 3-D structure of the frontal surfaces. We would also like to point the reader to supplementary video 3 (Beckert et al., 2022c). We consider interactive use of the presented method as a key aspect of 3-D analysis, the video provides an impression of the additional benefit gained through interaction. The 3-D depiction in Figure 7a reveals further frontal structures such as the large-scale frontal surface in the north (marked with a black arrow in Figure 7b), which is located above the 850 hPa level and could easily be missed in a 2-D analysis. Not

missing such potentially interesting structures is a key benefit of 3-D front detection compared to 2-D detection. We use $\theta_w$ as thermal input variable, which includes contributions from both temperature and humidity. It hence might be of interest to distinguish between fronts dominated by humidity or temperature. To do this, additional normal curve filters can be used. Fronts dominated by humidity are expected to have a much smaller temperature gradient across the frontal zone, hence, by adding an additional filter that evaluates the change of dry potential temperature $\theta$ allows us to discard features with an only

small $\theta$ gradient. Figure 7c-d shows temperature-dominated fronts, obtained by applying the additional normal curve filter of





$\theta$ with a fuzzy threshold interval of 0.6 – 1.0 K (100 km)$^{-1}$, the same value range used for $\theta_w$ (cf. Figure 2). This filter discards all humidity-dominated fronts. Note that the interactive adjustment of filter is also illustrated in the supplementary video 3 (Beckert et al., 2022c). The blue circle in Figure **7**c highlights an area of the cold front – note how upper-level parts (lighter green, towards the south) are discarded when humidity contribution is filtered. The vertical cross-section in Figure **7**d shows

$\theta$ and $|\nabla_h\theta|$, with the black arrow pointing at the area of the filtered out upper-level humidity-dominated front. The vertical cross-section also shows no temperature gradients, consistent with the interpretation that this is a humidity-dominated front. In Figure 7e-f a normal curve filter using specific humidity filter is applied instead, shifting focus to humidity contribution and discarding temperature-dominated gradients in $\theta_w$. In other words, temperature-dominated fronts are filtered out. The black circle in Figure 7e marks an area where a large-scale upper-level front is almost entirely discarded.

Finally, Figure 7g shows the impact of decreasing the smoothing length scale from 100 km to 30 km. This reveals frontal features on a different length scale. However, without adjusting the filter thresholds, the resulting fronts become cluttered. Figure 7f shows the same fronts as in Figure 7e but with adapted filter thresholds to compensate for the reduced horizontal smoothing length scale. Due to reduced smoothing, the smoothness of the frontal surfaces is reduced. Especially at the cold front, fluctuations in $\theta_w$ cause less continuous fronts (red circle). In addition, the reduced smoothing reveals other frontal

features on smaller scales, for example, the wrap-up of the occluded front around the cyclone center is more pronounced (orange arrow). Our recommendations for appropriate filter parameter intervals for different smoothing scales are summarized in Table 1 and are used throughout the paper, except where noted.

Table 1: Fuzzy frontal filter threshold recommendations for different smoothing length scales.

| Smoothing length scale km | TFP (K (100 km)$^{-2}$) | Frontal strength $\|\nabla_h\theta_w\|$ and $\|\nabla_h\theta\|$ (K (100 km)$^{-1}$) | Scale of detected frontal features |
|---|---|---|---|
| 100 | 0.2 – 0.4 | 0.6 – 1.0 | Large-scale |
| 50 | 0.4 – 0.8 | 1.0 – 1.6 | Large- and mid-scale |
| 30 | 1.5 – 2.5 | 1.2 – 2.2 | Mid- and small-scale |


**Figure 7. From 2-D to 3-D objective fronts.** Same data as in Figure 2 (*Friederike*) but showing the full 3-D structure of frontal surfaces in the lower and middle atmosphere. (a) 850 hPa frontal lines from Figure 2h with 3-D frontal surfaces between surface and 500 hPa, viewed from the top. (b) Same as (a) but from a tilted viewpoint looking north. Black arrow points towards upper front, which is missing in 850 hPa 2-D fronts. (c) Same as (b) but with additional fuzzy normal curve filter of $\theta$ between 0.6 - 1 K (100 km)$^{-1}$. This filter criterion removes purely humidity-based fronts such as the upper cold front (blue circle). (d) Same as (c) from a western point of view and vertical cross section shows $\theta$ and $|\nabla_h\theta|$. Blue arrow points to the filtered out upper cold front. (e) Same as (b) but with additional fuzzy normal curve filter of specific humidity between 0.1 - 0.2 g (kg 100 km)$^{-1}$. This filter criterion removes purely temperature-based fronts such as the northern upper cold front (black circle). (f) Same as (e) from a western point of view and vertical cross section shows $q$ and $|\nabla_h q|$. Back arrow points to the filtered out northern upper cold front. (g) Input field smoothed to a horizontal length scale of 30 km with same filtering applied as in (a). Note that with unchanged filter settings, the fronts appear cluttered due to small-scale details. (h) Same as (g) but with adapted filter settings for TFP between 1.5 - 2.5 K (100 km)$^{-2}$ and frontal strength between 1.2 - 2.2 K (100 km)$^{-1}$. Small-scale processes become visible, the cold front is less continuous (red circle), and the front wraps-up more around the cyclone centre (orange arrow).





## 5   Front detection applications

We illustrate how meteorological analyses can be performed using 2-D and 3-D front detection by investigating the two case studies introduced in Section 3. We first consider the validation of two conceptual models: WCB ascent in the vicinity of fronts (Section 5.1) and the development stages of a Shapiro Keyser cyclone (Section 5.2). Second, we focus on dynamical processes and frontal structures. We investigate how convection-permitting NWP simulations alter frontal surfaces compared to simulations in which convection is parameterized (Section 5.3). Finally, we compare our results to fronts analyzed by the UK Met Office and discuss the detection of secondary fronts as often shown in surface analysis charts of the UK Met Office (Section 5.4).

### 5.1   Validation of conceptual model: 3-D fronts and warm conveyor belt

Conceptual models and simplified illustrations are frequently used to explain the relation and dynamics of fronts and the WCB. Figure 8 shows an example of such an illustration in 2-D, a more sophisticated 3-D representation can be found, e.g., in Martínez-Alvarado et al. (2014, their Fig. 1). However, subsequent studies of these 3-D atmospheric features are usually conducted by means of horizontal or vertical 2-D slices through NWP data and it is less common to use a 3-D representation of 3-D atmospheric features (Rautenhaus et al., 2018). In this section, we demonstrate the use of 3-D front detection to visualize such conceptual models against NWP data by directly representing these features in 3-D.

Figure 9a-c shows the evolution of 3-D fronts from 03:00 UTC to 09:00 UTC 23 September 2016 of *Vladiana*, together with a selection of WCB trajectories that ascend at the selected times. During this period the frontal system moves eastwards. At 03:00 UTC the selected WCB trajectories are located in the lower troposphere near the surface in the warm sector and move along the cold front in a north-easterly direction (Figure 9a). At 06:00 UTC most of the WCB trajectories are in their ascent phase (Figure 9b, d-f), and at 09:00 UTC the majority of the WCB trajectories have risen above 500 hPa (Figure 9c). The selected trajectories choose different pathways for their ascent: some rise directly at or ahead of the cold front, and others rise above the warm front. While trajectories rapidly increase in altitude when lifted spontaneously at the cold front, trajectories at the warm front ascent more slowly and gradually. In Figure 9d-f the difference between cold frontal and warm frontal ascent is emphasized. It shows frontal surfaces at 06:00 UTC together with 48-h WCB trajectories, which are coloured according to their maximum ascent rate within 2 h. The WCB trajectories are displayed thicker between 03:00 UTC and 09:00 UTC. Most of WCB trajectories at the cold front have a maximum ascent rate faster than 300 hPa within 2 h. In contrast, trajectories at the warm or occluded front ascend more slowly, with maximum ascent rates below 200 hPa in 2 h. In the upper troposphere, the WCB splits into two outflow branches, a cyclonic branch which turns westward and an anticyclonic branch which turns eastwards. WCB trajectories ascending ahead of the cold front tend to take the anticyclonic outflow, while warm or occluded frontal WCB trajectories tend to take the cyclonic outflow. We hypothesize that trajectories that rapidly ascend at the cold front experience jet wind speeds earlier following the anticyclonically turning jet stream and are thus deflected into the downstream ridge (see Figure A 2). The 3-D visualization corroborates the conceptual model of how WCB ascent relates to

fronts, and highlights the presence of smaller-scale convective ascent structures embedded in the WCB discussed in recent studies (cf. Rasp et al., 2016; Oertel et al., 2019, 2020; Blanchard et al., 2020). The 3D visualization of rapidly and more slowly ascending high-resolution WCB trajectories further shows their similarity to the so-called 'escalator-elevator' concept

of WCB-embedded convection which was proposed by Neiman et al. (1993) to distinguish between fast ascent and more gradual frontal upglide. By looking at the 3D structure of the trajectories, this concept appears suitable.

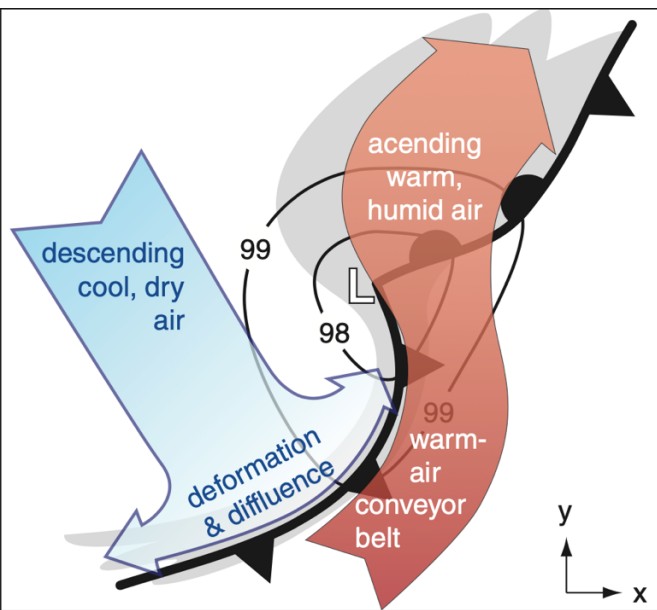

**Figure 8: Conceptual model of fronts and WCB showing large scale ascending and descending air in the vicinity of an extratropical**
**cyclone (Figure reproduced from Stull, 2017, © Stull, 2017, CC-NC-SA 4.0 license).**







**Figure 9. (a-c)** Temporal evolution of 3-D frontal structures and WCB trajectories of *Vladiana* on 23 September 2016. **(d-f)** Same time as (b) but visualized from different viewpoints (cf. Figure 3). WCB trajectories starting on 22 September 2016, 18:00 UTC and 23 September 2016, 0:00 UTC and displayed for a period of 48 h. Trajectories are plotted thicker during the period between 03:00 UTC and 09:00 UTC and are coloured according to their maximum 2 h ascent rate. For the full temporal development of 3-D frontal structures, jet stream and WCB trajectories see the supplementary video Beckert et al. (2022a).



## 5.2   Validation of conceptual model: 3-D Shapiro-Keyser Cyclone

Figure 10 extends the 2-D frontal analysis of *Friederike* shown in Figure 4 and shows the temporal development of the 3-D structure. In 3-D, the typical characteristics of a Shapiro-Keyser cyclone (Shapiro and Keyser, 1990) with its distinctive frontal

T-bone structure and the four cyclone stages can be well observed. However, at different elevations the four stages, as described in Schultz and Vaughan (2011), occur at different times:

- Red and orange front: Stage I, incipient frontal cyclone. A perturbation of the frontal structure is already present in the upper atmosphere. This disturbance will later develop into the frontal wave. However, the frontal surface in the lower atmosphere is unperturbed.

- Orange, yellow, green front: Stage II, frontal fracture. The timing of frontal fracture strongly depends on the vertical level. In the lower troposphere the cold front is separating from the main front. In the upper troposphere, a connection between the cold front and the main part of the frontal surface still exists.

- Green and blue front: Stage III, bent-back warm front, and frontal T-bone structure. At lower levels, the cold front lies almost perpendicular to the warm front, showing the typical Shapiro-Keyser T-bone structure. Interestingly, the

upper part of the cold front also bends slightly towards the south, following the lower part of the cold front, but a connection to the warm front remains.

- Blue and purple front: Stage IV: warm-core frontal seclusion. The warm front wraps-up around the warm air near the cyclone centre. The separated lower part of the cold front moves further south and the upper cold front dissipates.

In this example, uniquely assigning the 3-D frontal structure at specific time steps to the Shapiro and Keyser stages is not

possible. As described, frontal evolution does not occur synchronously at all elevations, creating a temporal offset of the stages at different elevations. We could also not find a height level where the 2-D fronts could be unique assigned (see Figure 4). It is important however that the 3-D front detection is capable of detecting all the characteristic structures of the Shapiro-Keyser model, even though a one-to-one assignment to the stages is not possible. Another example of the 3-D frontal development with typical characteristics of a Shapiro-Keyser cyclone, storm *Egon* (11-13 January 2017; Eisenstein et al. (2020)), is shown

in Figure A 3 in the appendix of this study. Again, the visual analysis shows that frontal evolution does not occur synchronously at all elevations, creating a temporal offset. For example, frontal fracture does not occur at all elevations simultaneously. The time step on 12 January 2017, 23:00 UTC shows the development of the bent-back warm front in upper levels, whereas the frontal fracture is not yet complete near the surface. These examples suggest a more nuanced view of the Shapiro-Keyser model, where there is a significant 3-D component to the evolution of a cyclone through the different stages of the conceptual

model.





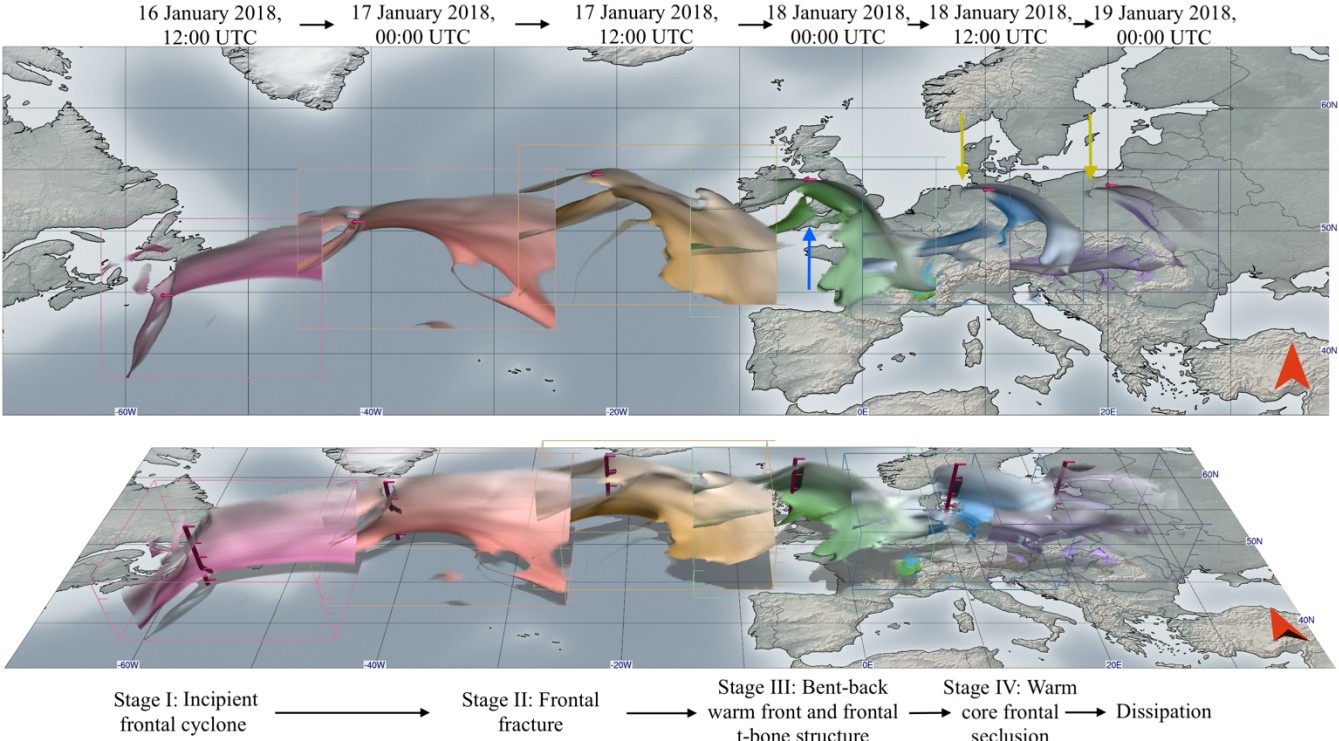

**Figure 10. Successive time steps of objective 3-D frontal structures showing the temporal development of *Friederike* (16 to 19 January 2018), as detected in ERA-5 reanalysis data. Blue arrow points towards frontal fracture and yellow arrows point towards warm core frontal seclusion.**





### 5.3 Convection-induced changes of the cold front


Here we compare fronts of convection-permitting NWP simulations with fronts in simulations where convection is parameterized using *Vladiana* as an example. We focus on the southern end of the cold front (green box in Figure 3a) where mid- and small-scale convection occurs in this WCB. Oertel et al. (2019) highlight (embedded) convection with lightning near the trailing edge of the cold front on 23 September 2016, 06:00 UTC. To detect mid-scale frontal features induced by

convection the input field, $\theta_w$ is smoothed to a horizontal length scale of 50 km (cf. Table 1). Figure 11 shows detected 2-D fronts at 850 hPa together with fronts of UK Met Office surface charts, at 700 hPa and at 500 hPa. The yellow dot at the southern end of the cold front marks the position of the observed embedded moist convection. The COSMO simulation shows strong ascending motion around the convection at all plotted vertical levels (Figure 11d-f). However, in the ECMWF data (Figure 11a-c) where convection is parameterized, the vertical velocity field shows no significant local maximum. The detected

cold front of both simulations follows the cold front of the UK Met Office surface analysis chart. However, in the vicinity of convection and at 850 hPa the cold front of the COSMO simulation brakes apart, while the cold front detected in ECMWF is a continuous line. At 700 hPa the cold front detected in ECMWF data is weak and broken while the cold front detected in COSMO data is a continuous line. At 500 hPa the cold front is shifted towards north and less continuous in the COSMO data compared to ECMWF data. 3-D fronts of ECMWF data in Figure 12a show a vertical gap in the frontal surface with an extent

of about 100 hPa. It is located at the southern end of the cold front approximately between 700 and 600 hPa and steadily ascends towards the north until it ends above 500 hPa. These kind of gaps in the cold front have been observed in other studies (Geerts et al., 2006) and are associated with weaker temperature gradients at this elevation. However, this gap is not distinctively mirrored in 3-D fronts detected in COSMO data (Figure 12b). Near the centre of the convective updraft (yellow pole in Figure 12b) the gap in the cold front is closed. We hypothesise that convection can impact the structure of the cold

front, especially in the vicinity of convective updrafts. The strong horizontal convergence associated with convective updrafts in this area (see Figure A 5) may act frontogenetically by strengthening the horizontal temperature gradients in the mid-troposphere, with the result that the upper and lower-level frontal surfaces are connected. The time evolution of the COSMO 3-D front (Figure A 4) suggests that the intensification of the mid-level cold front is a transient feature that occurs at the time of convection and disappears as soon as the convection weakens again. In simulations where convection is parameterized,

however, the convection scheme may not activate at that time and location, and additionally the feedback of the convection scheme on the grid-scale variables may differ from their explicit model representation (as shown in this example). We propose that future detailed comparisons of the 3-D frontal structure in simulations with explicit vs. parameterized convection may improve the understanding of processes that strengthen the cold front.



**Figure 11.** Impact of convection on detected frontal structure on 23 September 2016, 06:00 UTC in green subarea shown in Figure 3a, comparing (top) ECMWF analysis at (a) 850 hPa, (b) 700 hPa, and (c) 500 hPa, to (bottom) COSMO analysis at (d) 850 hPa, (e) 700 hPa, and (f) 500 hPa. Maps show objective 2-D fronts (blue tubes), UK Met Office fronts (red tubes), $\theta_w$ (colour), $|\nabla_h \theta_w|$ (grey shades), upward air velocity (contour lines; orange=upwards, black=zero, green=downwards, contour line spacing of 0.02 m s$^{-1}$).

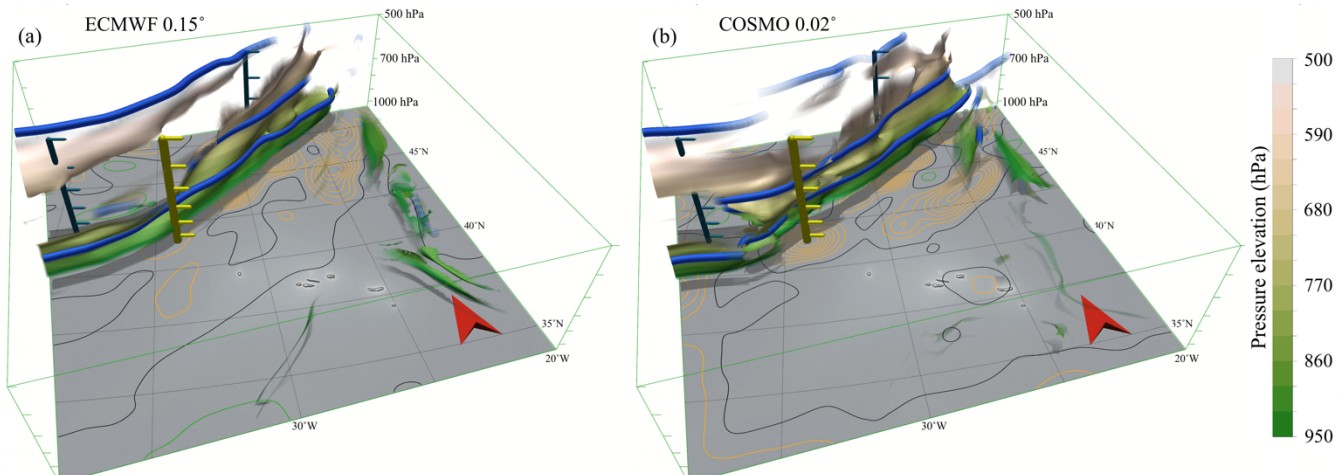

**Figure 12. 3-D view of the 2-D frontal structures shown in Figure 11. 2-D objective fronts (blue tubes) at 850 hPa, 700 hPa, and 500 hPa (cf. Figure 11) in the context of full 3-D frontal structures. Contour lines show upward air velocity at 700 hPa (orange=upwards, black=zero, green=downwards, contour line spacing of 0.02 m s⁻¹). (a) ECMWF analysis and (b) COSMO analysis. Same data and region as in Figure 3. The yellow pole marks the centre of the convective updraft (cf. Figure 11), the red arrow points northward.**

### 5.4 Secondary fronts

Secondary fronts are commonly analysed by the UK Met Office and seen in their surface analysis charts. Beside other variables, the UK Met Office uses the wet-bulb potential temperature as primary thermal variable for their front detection in surface analysis charts (N. Armstrong, UK Met Office, pers. comm., 2022). In this section, we consider a secondary front which occurs ahead of the warm front of *Friederike*. We investigate if the front detection algorithm can detect such secondary fronts and how secondary fronts depend on the detection variable. Red tubes in Figure 13 show the positions of fronts analysed by the UK Met Office for 18 January 2018, 12:00 UTC. The most easterly front, ranging from northeast Italy up to the southern border of Denmark, is a typical secondary warm front as often analysed by the UK Met Office. Figure 13b shows fronts detected in $\theta_w$ at 850 hPa (green tubes). In general, the structure of fronts detected in $\theta_w$ agrees well with fronts of the UK Met Office, despite some smaller differences. In particular, the secondary front is shorter in its horizontal extent and the wrap-up of the occluded front around the cyclone centre is more pronounced. Figure 13c shows fronts detected in $\theta$ at 850 hPa (green tubes). There is no indication for secondary fronts in this analysis, as no strong horizontal gradients of $\theta$ are present in this area. Hence, the presence of the secondary front detected by $\theta_w$ results from moisture gradients. Furthermore, the structure of the primary fronts is less continuous and deviates more from the UK Met Office analysis. Figure 14 shows the 3-D frontal surfaces of $\theta_w$ (Figure 14a) and $\theta$ (Figure 14b). The 3-D frontal structure illustrates that the secondary front detected in $\theta_w$ is a shallow atmospheric feature and only present in the lower troposphere at around 850 hPa. However, above 700 hPa another secondary front is detected, which is present in both thermal variables, $\theta_w$ and $\theta$ and extends at least up to 500 hPa. For this case study we conclude that the lower atmospheric secondary front is a moisture feature, and thus, can only be detected in a variable that includes humidity formation. Furthermore, $\theta_w$ as detection variable results in more continuous fronts compared



to $\theta$. We again would like to point the reader to the supplementary video 3 (Beckert et al., 2022c), which illustrates the benefit of interactive exploration and analysis of the detected fronts within Met.3D.

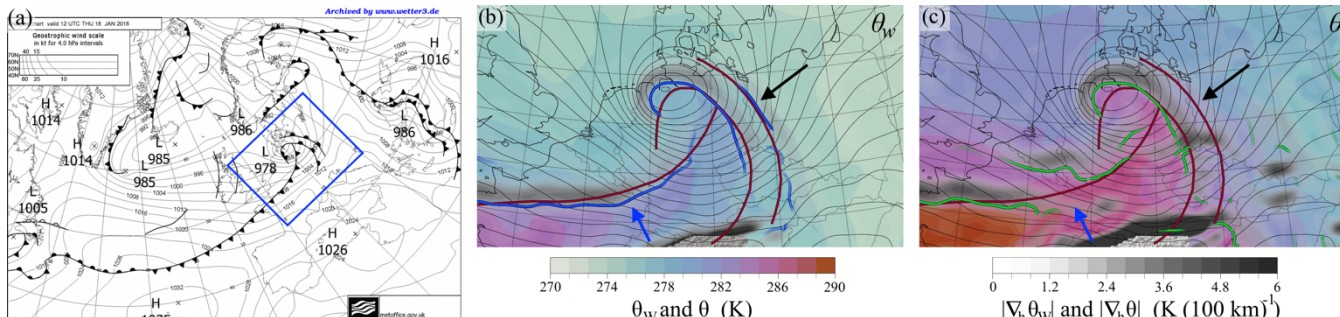


**Figure 13. Comparison of UK Met Office fronts with objective fronts, of *Friederike* on 18 January 2018, 12:00 UTC. (a) UK Met Office surface analysis chart. Blue box marks analysed area. (b) Objective 850 hPa 2-D fronts (blue lines) as detected from ECMWF HRES $\theta_w$ (colour; grey shading shows $|\nabla_h \theta_w|$), UK Met Office fronts (red lines), and mean sea level pressure (black contour lines). (c) Same as (b) but objective 2-D fronts (green lines) based on $\theta$. Note that the secondary front (black arrow) is only detected when**


**using $\theta_w$. When based on $\theta$, the cold front (blue arrow) breaks up and is less continuous compared to the cold front based on $\theta_w$.**

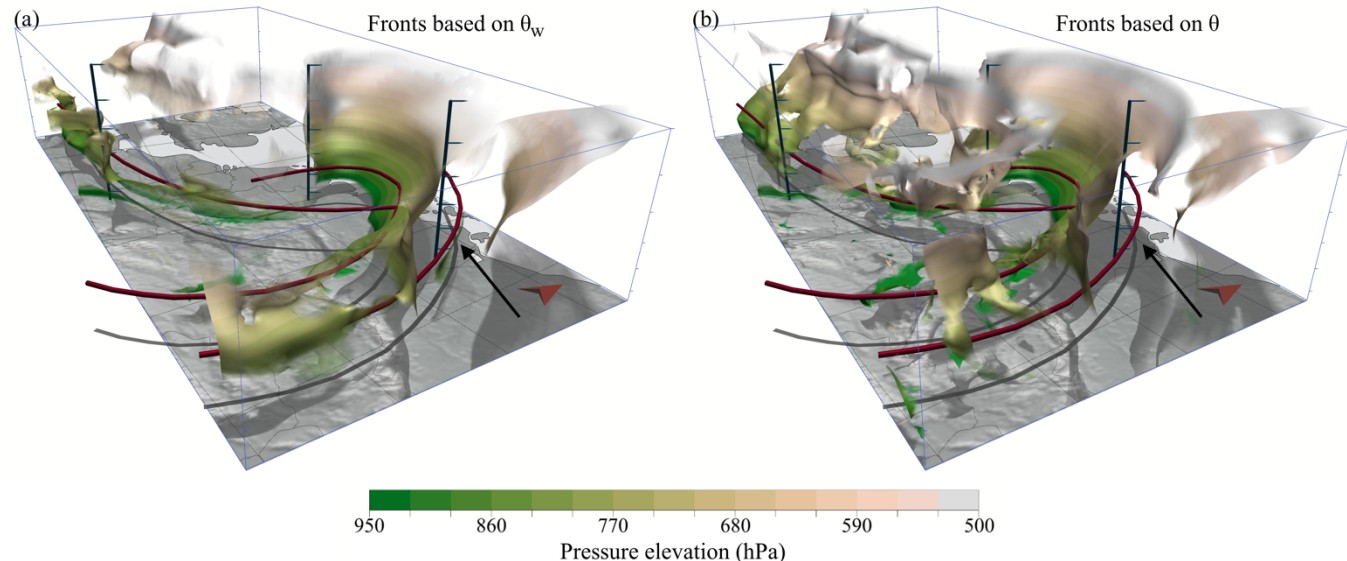

**Figure 14. 3-D view of Figure 13b and c. Red tubes show UK Met Office fronts, 3-D objective fronts are coloured according to pressure elevation. (a) Objective fronts based on $\theta_w$ and (b) based on $\theta$. Note that the secondary front (black arrow) is a feature of $\theta_w$ and only occurs around 850 hPa. For an easier grasp of the 3-D perspective see Beckert et al. (2022).**






## 6 Summary and discussion

This article explores how objective 2-D and 3-D front detection and visualization, integrated into an interactive 3-D visual analysis environment for atmosphere data, can be used to study frontal dynamics within mid-latitude cyclones and thus be beneficial for weather forecasting and research. The presented method builds on approaches previously introduced by Hewson

(1998) and Kern et al. (2019) and is applicable to gridded data from state-of-the-art NWP models. It facilitates rapid analysis of 3-D frontal dynamics, including objective comparisons of detected frontal structures between datasets from different numerical models or ensemble members, also at different model resolutions. We addressed the objectives of (a) identifying appropriate detection parameters including data smoothing and filtering thresholds, to ensure objective comparability, and (b) evaluating the benefit of 3-D IVA of frontal surfaces through case study investigations, including interpretation based on

conceptual models, and comparison of frontal structures between different numerical models and with manually produced surface analysis charts.

We find that the integration of 3-D front detection with 3-D IVA (in our case in the open-source meteorological visual analysis framework Met.3D) facilitates rapid analysis of complex weather situations, in part because the detected fronts can be visualized jointly with interactively placed depictions of other meteorological quantities.

The choice of the thermal variable is essential for the presented approach. We show that $\theta_w$ is most suitable since, in contrast to $\theta$, it considers reversible moist processes in the atmosphere. The resulting fronts are longer and more continuous. A disadvantage of $\theta_w$ is that it also detects purely humidity-based fronts. Separately filtering frontal feature candidates according to humidity and $\theta$ gradients, however, allows us to distinguish humidity-based from temperature-based fronts. The choice of filter parameters and filter thresholds to obtain meaningful frontal structures is challenging. These settings depend on the

thermal input variable's horizontal smoothing length scale, which determines the "spatial scales" of detected frontal features (large-scale smoothing of the thermal input field results in detection of large-scale frontal features and vice versa). The distribution of gradient magnitudes shows that different smoothing length scales require different filter thresholds to obtain meaningful fronts. Large-scale smoothing requires less restrictive filter thresholds compared to small-scale smoothing. We present recommendations for future users on how to tune filter thresholds according to the previously applied smoothing length

scale (Table 1).

The application of the proposed approach to case studies of midlatitude cyclones provides detailed information about the temporal evolution of 3-D front characteristics. We demonstrate the use of 3-D front detection to visualize dynamic relations of features in the context of fronts in NWP data by directly representing these features in 3-D. In a case study of cyclone *Vladiana* (September 2016) we evaluate the conceptual model of the WCB. At the cold front, WCB trajectories ascent fast,

experience jet wind speeds early and follow the anticyclonically turning jet stream. In contrast, WCB trajectories ascending at the warm or occluded front show a slower ascent rate and tend to take the cyclonic outflow branch in the upper troposphere. These observations agree well with conceptual models of fronts and WCB as proposed in literature. In a second case study of cyclone *Friederike* (January 2018), we visually analyse the 3-D temporal evolution of fronts in a Shapiro-Keyser cyclone and





compare our results to the conceptual model proposed in the literature. We observe that the frontal evolution of the Shapiro-Keyser cyclone does not occur synchronously at all elevations. However, all characteristic stages of the conceptual model of the Shapiro-Keyser cyclone could be observed in NWP data. Our next example considers the influence of convection at a cold front. We find that cold frontal convection can influence the 3-D frontal structure and show that for *Vladiana* convection likely acts frontogenetically and strengthens the cold front in the mid-troposphere temporarily. Finally, we compare the objective 3-D frontal structures with 2-D fronts in UK Met Office surface analysis charts and investigate the occurrence of secondary fronts often present in UK Met Office surface analyses. The objective 3-D fronts are consistent with the UK Met Office fronts if $\theta_w$ is used for front detection. This is no coincidence as $\theta_w$ is the primary thermal variable used for the manual front detection by the UK Met Office. For *Friederike*, we show that the secondary front corresponds to a humidity-dominated rather than a temperature-dominated front.

Alternative front detection methods have been described in the literature. A 2-D front detection approach also applicable to kilometre-scale resolution data was proposed by Jenkner et al. (2010). Because of high sensitivity to local noise in higher derivatives, their approach uses the zero lines of the TFP (second derivative) as frontal candidates, which correspond to the steepest gradient within the frontal zone. However, this does not match the most common definition of a front as the boundary of the frontal zone located on the warm air side (cf. Renard and Clarke, 1965). We argue that an advantage of our approach in particular for case studies is that also in kilometre-scale data fronts are detected at this warm air side, albeit at the cost of potential smoothing artefacts.

An entirely different approach using artificial neural networks to detect 2-D fronts was recently proposed by Niebler et al. (2021). However, their approach has requirements that may be disadvantageous. First, the neural network is trained on a specific data set and region and is not readily transferable to other regions, data sets and data resolutions. Second, to train the neural network a large training, testing and evaluation data set with predefined fronts as ground truth is necessary. Niebler et al. (2022) use predefined fronts from several weather services which are manually or semi-automatically detected as ground truth. It is debatable how objective the results are when training the neural network with subjectively analysed fronts as ground truth.

The front detection and visualization approach presented here has the potential to be used operationally. Being integrated in Met.3D, other meteorological variables can be analysed in conjunction with the 3-D frontal structures. This facilitates the rapid analysis of complex weather situations, as required in operational settings (cf. Rautenhaus et al., 2018). Further fields of application include the feature-based analysis of forecast uncertainty represented by ensembles simulations (albeit comparative visualization of features from many ensemble members will be challenging), climatological studies of frontal characteristics derived from the 3-D features, and investigation of the relation of frontal structures to other physically meaningful features in the 3-D atmosphere, including the jet stream – this will be beneficial for studies that contribute to the understanding of complex dynamical processes in the atmosphere.



**Code availability**

The code of the open-source visualization framework Met.3D is available at https://gitlab.com/wxmetvis/met.3d (Met.3D – Code Repository, 2022). User and developer documentations including further resources are available at
https://met3d.wavestoweather.de (Met.3D – Homepage: Interactive 3D visualization of meteorological simulations, 2022) and https://collaboration.cen.uni-hamburg.de/display/Met3D/ (Met.3D – Documentation: User Documentation, 2022).

**Video supplement**

The following movies illustrate interactive visual data analysis using Met.3D and provide supplementary insights into the 3-D
dynamics of frontal structures, jet stream and WCB trajectories, and illustrate the benefit gained from interactive use of 3-D visual analysis.

–    Video 1: Development of 3-D frontal structures, jet stream and WCB trajectories of Vladiana (Beckert et al., 2022a).
–    Video 2: Video 2: Comparison of objectively detected 3-D fronts in wet-bulb potential temperature and potential temperature of Friederike on 18 January 2018, 12:00 (Beckert et al., 2022b).
–    Video 3: Interactive front analysis of storm Friederike using the open-source meteorological 3-D visualization framework "Met. 3D" (Beckert et al., 2022c).

**Competing interests**

The authors declare that they have no conflict of interest.

**Author contributions**

AB designed and implemented the algorithm, performed the analysis, created the visualizations, and wrote the manuscript. LE contributed to the meteorological related results, especially to the characterisation and discussion of the Shapiro-Keyser cyclone. AO contributed to the meteorological related results, especially to the characterisation and discussion of the WCB.
TH contributed to the design of the algorithm and secondary front discussion. MR and GC proposed, supervised, and administrated the study. All authors contributed to writing and revising the manuscript.

**Acknowledgement**

This research leading to these results has been done within subprojects C9 (AB, GC, MR), C5 (LE), and B8 (AO) of the
Transregional Collaborative Research Center SFB/TRR165 "Waves to Weather" (www.wavestoweather.de) funded by the German Research Foundation (DFG). The authors would like to thank the Institute for Atmospheric and Climate Science at ETH Zürich for providing the high-resolution COSMO dataset of *Vladiana*.





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





## Appendix A

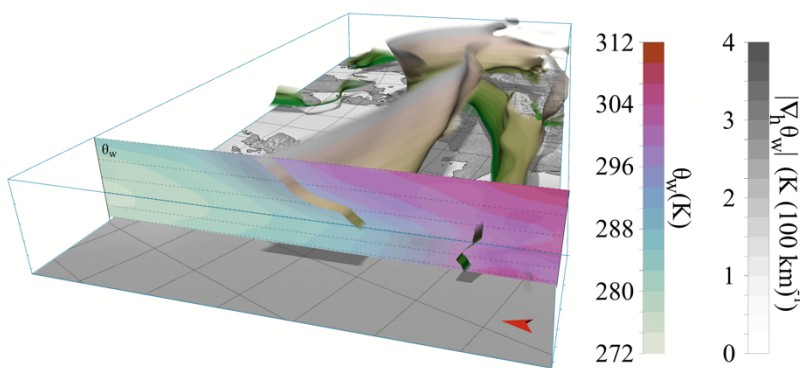

**Figure A 1: Same as Figure 7b but from a western point of view looking eastward. Vertical cross section shows $\theta_w$ and $|\nabla_h \theta_w|$.**

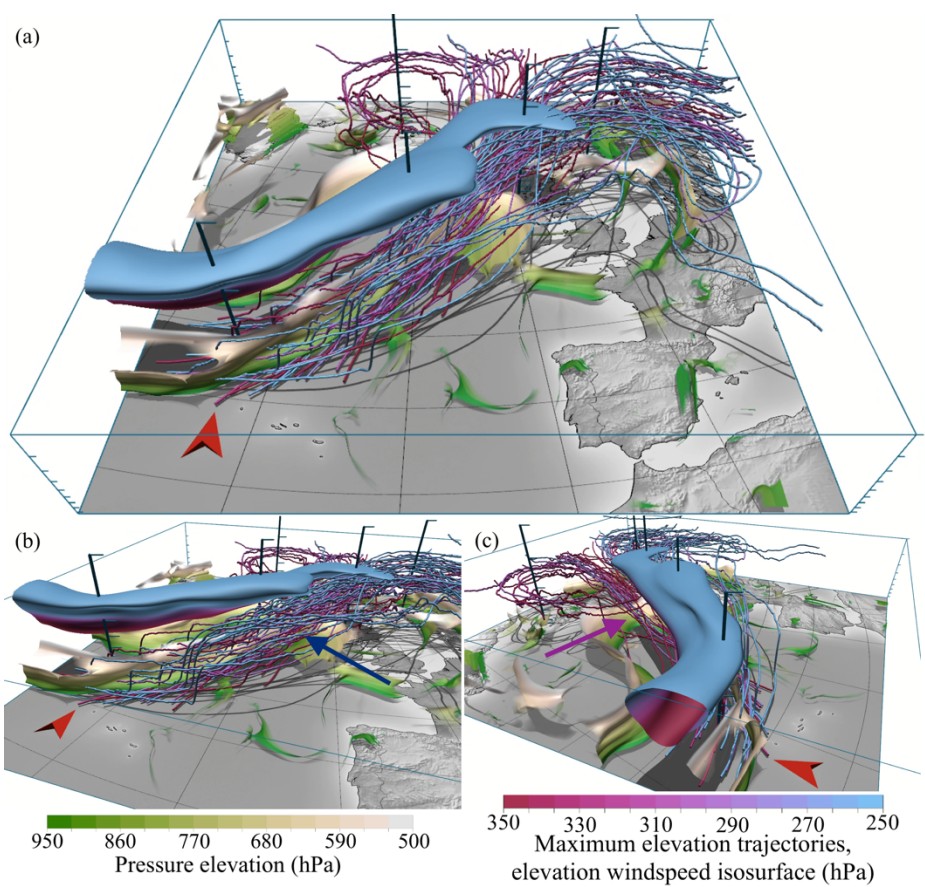

**Figure A 2. Different viewpoints on 3-D frontal structures and jet stream (isosurface of 50 ms⁻¹ windspeed) of *Vladiana* on 23 September 2016, 09:00 UTC and WCB trajectories. WCB trajectories are coloured according to their maximum height. Fast cold frontal ascending WCB trajectories tend to ascent towards heigher altitudes and following the jet stream anticyclonic outflow branch (blue arrow in b). Warm frontal ascending WCB trajectories tend to ascend towards lower altitudes and following the cyclonic outflow branch (purple arrow in c). For the full temporal development of this scene see the supplementary video Beckert et al. (2022a).**






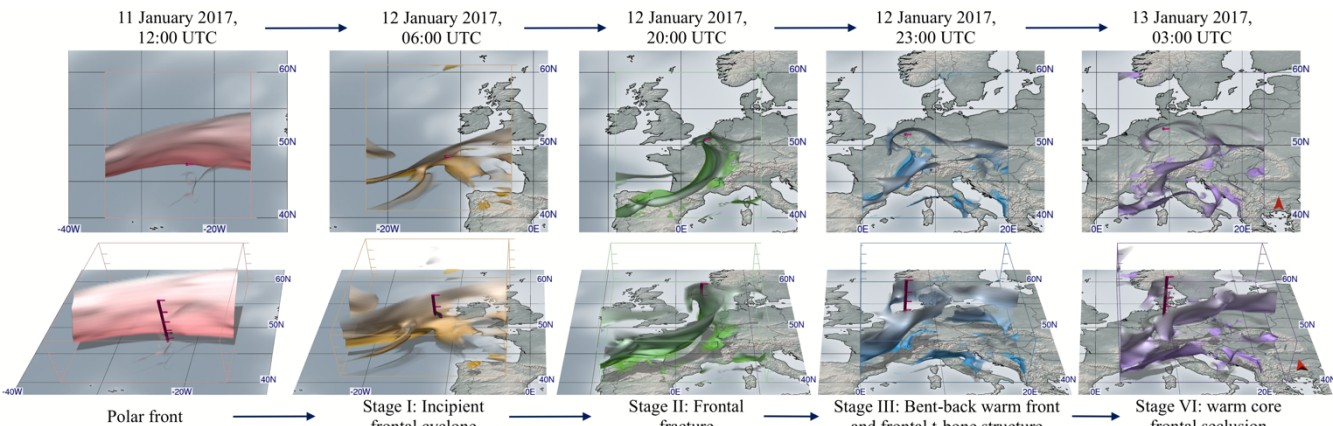


**Figure A 3. Successive time steps of objective 3-D frontal structures showing the temporal development of storm *Egon* (11 January 2017 12:00 UTC to 13 January 2017, 03:00 UTC), as detected in ERA-5 reanalysis data.**







**Figure A 4. 3-D development of frontal structures of the 3-D fronts shown in Fig. 12. Contour lines projected onto the surface show**
**upward air velocity at 700 hPa (orange=upwards, black=zero, green=downwards, contour line spacing of 0.02 m s⁻¹). Right panel shows ECMWF analysis and left panel shows COSMO analysis. The yellow pole marks the centre of the convective updraft at 06:00 UTC, red arrow points northward.**





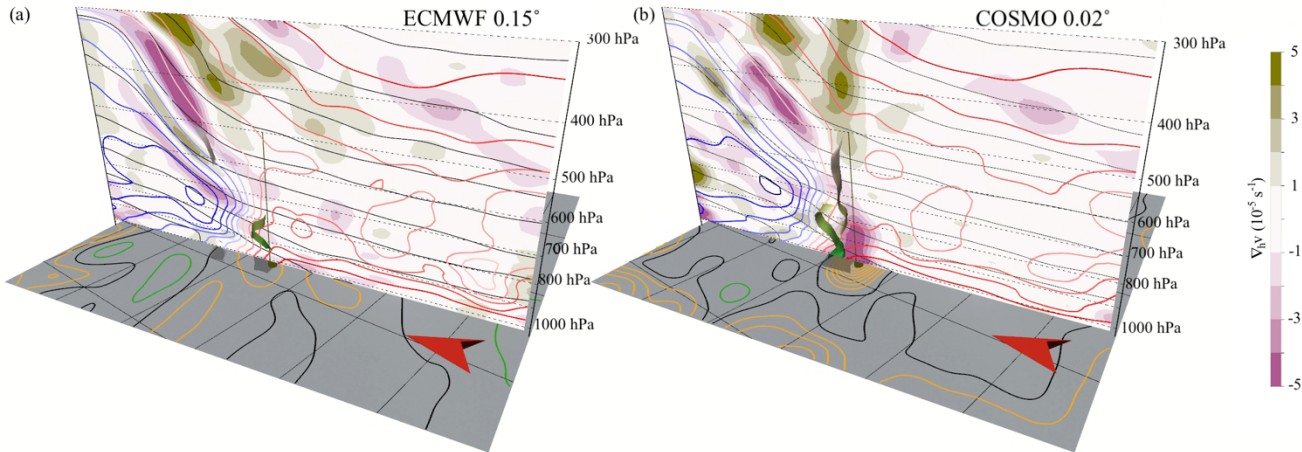

**Figure A 5. Case *Vladiana* and same time and area as in Figure 12. 3-D fronts and vertical cross-section of wind divergence (colour), $\theta_w$ (coloured contour lines, spacing 1 K), and $\theta$ (black contour lines, spacing 5 K). Upward air velocity contour lines at 700 hPa are projected onto the surface (orange=upwards, black=zero, green=downwards, contour line spacing of 0.02 m s$^{-1}$). Right panel shows ECMWF analysis and left panel shows COSMO analysis.**
