# Peer review of "The three-dimensional structure of fronts in mid-latitude weather systems as represented by numerical weather prediction models"

_Weather and Climate Dynamics, 2022_

## Author Comment (AC1)

**RESPONSE TO REVIEWERS**

**The three-dimensional structure of fronts in mid-latitude weather systems as represented by numerical weather prediction models**

We would like to thank both reviewers and the editor for their time spent on reviewing our manuscript. For our resubmission to GMD, we have revised the article according to the provided comments. Below, we provide point-by-point replies (blue color) to each comment (black and in italics).

**Reviewer #1:**

We already responded to Reviewer #1 in a separate reply in the interactive discussion (see https://doi.org/10.5194/wcd-2022-36-CC1). Below, we provide additional replies.

*(1) The front surface detection seems to be only a minor modification/optimisation of the algorithm introduced and implemented in Kern et al. (2019). Judging from the illustrations in Kern et al. (2019), the algorithm was already then implemented in Met.3D. Yet, the algorithm is introduced and discussed here in as much detail as if it was new. Further, the authors "validate" well-established meteorological concepts such as the Shapiro-Keyser cyclone model and the spatial relation between WCBs and fronts using their visualisation and front detection algorithm. Given how successful these concepts have been over decades, I find this quite assuming. If these concepts had failed to show up in their analysis, I would much rather doubt the implementation and visualisation in question rather than these meteorological concepts. Now, given that everything looks as expected, I am unsure what to take away from the "validation" beyond that the algorithm and visualisation is working fine---and so much that had already been shown by Kern et al. (2019).*

Response:
Thank you for your comment. As we have stated in our previous reply, there were two major objectives that we wanted to achieve with our work: (a) a reimplementation and generalization of the Kern et al. (2019) method such that it is robust when used with current model data and can handle additional filter options, and to document the tool and make it available to the community as an open-source release, alongside the paper (which the original Kern et al. method was not); and (b) provide guidance to researchers on how to use 3-D front detection and visualization by investigating suitable method parameters and by showing the potential of the method for meteorological analyses with selected examples (which had not been done before).
We agree, however, that our choice of words could have led to misunderstanding of our objectives.

Action:
- To avoid confusion about what we intended to say by "validate", we rephrased the manuscript parts on "validation of conceptual models". In particular, we changed "evaluate" / "validate" to "examine conceptual model in 3-D by means of NWP data"

*(2) The authors discuss briefly the best choice of thermodynamic variable for the front detection. This choice remains an subject of debate, and a new perspective on this choice could warrant another publication. This would however require considerable additonal analyses; based on only two case studies, the authors are not in a position to give general recommendations (as presently done in the summary and discussion section).*

Response:
Thank you for your comment. We agree that the limits of our analysis on which thermal variable is best suited for 3-D front detection should be more precisely stated.

Action:
- We rephrased the according paragraphs to clarify that this analysis may not be readily transferable to other case studies.

*(3) Similarly, a front classification into humidity and temperature-dominanted fronts would most likely be worthwhile and well warrant a publication. But this aspect is discussed by far too superficially to justify the publication of the present manuscript.*

*(4) Similarly, the comparison of WCBs and frontal structures in parameterised-convection versus convection-resolving models is both timely and interesting. It would certainly warrant a publication of its own. But again, this aspect is discussed by far too superficially here.*

Response:
Thank you for your comments. We agree that a detailed analysis of these two points would be of interest and a worthwhile effort. However, we decided that for our current study, such detailed analysis goes beyond our intended scope – which led to our decision to resubmit the manuscript to GMD.

**Reviewer #2:**

**Major points:**
*RC2: The paper feels disjointed in its current setup. I feel that Section 3 should be re-distributed into Section 5 so that 3.1 gets mixed in with the introductory paragraph of section 5, 3.2 goes at the start of section 5.1 (5.1.1 and 5.1.2?) and section 3.3 does likewise in section 5.2 (5.2.1 and 5.2.2). There would then be a nice flow from introducing the case study and the plots would flow logically from "concept" to "analysis". Currently there are methods in section 2, an introduction of the case studies in section 3, another methods section is given in section 4 and we then return to the case studies in section 5. As you can see, the paper "jumps" around a bit. Putting section 3 into section 5 and re- ordering the figures will make it flow much better instead of having to refer-back to section 3 from section 5. Furthermore, section 5 jumps around too – better to stick with the one case cyclone and focus in on several features associated with it i.e. make 5.3 part of 5.1 e.g. 5.1.1, 5.1.2 and 5.1.3 then push section 5.4 into 5.2 (5.2.1, 5.2.2 and 5.2.3).*

Response:
Thank you for your comment on the structure of the manuscript. We agree that your suggestion improves the structure of the manuscript.

Action:
- We adapted the structure of the paper to join the case studies, meteorological case description and the data section into a single section (new Sect. 4).
- The old Sect. 3 has been merged with the old Sect. 5. The old Sects. 4, 5, 6 have been renumbered Sects. 3, 4, 5 in the revised manuscript to be submitted to GMD.
- The new Sect. 4 (old Sect. 5) has been restructured:
  - First, we introduce the meteorological theory (Sect. 4.1, former Sect. 3.1).
  - Second, we describe the meteorological situation and introduce the data sets used for the case *Vladiana* (Sect. 4.2, former Sect. 3.2) followed by the two case analyses (Sect. 4.2.1, former Sect. 5.1 and Sect. 4.2.2, former Sect. 5.3).
  - Third, we describe the meteorological situation and introduce the data sets used for the case *Friederike* (Sect. 4.3, former Sect. 3.3) followed by the two case analyses (Sect. 4.2.1, former Sect. 5.2 and Sect. 4.2.2, former Sect. 5.4).

*RC2: Section 2.3 and Figure 2: The wording in the numbered list should match the plots and the plots should then be referred to in each of the points of the numbered list e.g. point 1 goes with Fig 2a, point 2 with Fig 2b etc. It seems that this Figure-numbered list relation does not hold true in all cases so the authors should either adjust the list or adjust the figure to make the two complimentary. There are also no descriptions of the panels in the Figure 2 caption so you should say "see Section 2.3 for a description of the panels a-h" in the figure caption.*

Response:
Yes, we agree. Thank you for your comment.

Action:
- We changed the manuscript according to your suggestion. We removed the third point in the numbered list in Sect. 2.3. The Figure and numbered list relation holds

true now. Beside that we added a reference to each item in the numbered list pointing to the associated subfigure.

*RC2: Figure 3b: This is difficult to interpret and would probably be better if the fronts were colour coded to show the difference between warm, cold and occluded (red, blue and purple?). I just find the figure to have a lot of "green swirls" that really need to be separated visually to make the features stand out. Furthermore, the scale is too smooth to really show the location of the fronts in the vertical. Using different colours for the type of front or markedly different colours at each pressure level might make these (and all the other figures that use the green-white colour scaling for the pressure heights) clearer.*

Response:
Thank you for your comment. We agree, this figure is hard to interpret in its current version, and the green-white colourbar can cause difficulties in interpreting the pressure elevation of the frontal surfaces.

Action:
- We re-coloured the fronts in Figure 3 (Fig. 6 in the new manuscript) according to warm- and cold air advection (following Hewson, 1998, Kern et. al. 2019).
- We reduced the vertical scaling and slightly adapted the viewing angle of Figure 3b to improve clarity in the spatial structure.
- Also, the green-white colourbar for pressure elevation is replaced by a more distinctive colourbar with reduced smoothness in subsequent figures.

*RC2: Figure 9 and Lines 475 – 490: This whole description is very difficult to see as both the paper and online plots are far too "busy". Surely you can distinguish between the "fast ascending" and "slow ascending" trajectories and plot them separately. I would discard figures 9e and 9f and replace them with figures showing "fast" and "slow" ascending trajectories. You would then only need to slightly re-word lines 475-490 to account for this change. I feel that the whole paragraph would then read much better with the adjusted figure.*

Response:
Thank you for your comment. We much appreciate your suggestion to separate the trajectories into "fast" and "slow" ascending trajectories.

Action:
- We replaced Figure 9d and 9e. Figure 9d shows "fast" (more than 200hPa within 2h) ascending WCB trajectories and Figure 9e shows slow (less than 200hPa within 2h) ascending trajectories now.
- Figure 9f shows all WCB trajectories as well as the jet stream isosurface.

*RC2: Figure 10: I'm not convinced this figure shows anything a 2D figure wouldn't show. It is trying to do too much by overlaying everything on the same plot and so the details are lost. I would like to see this separated out into ~6 panels that show the 3D structure clearly from the best angle. I would also suggest removing the land and just focussing on the cyclone itself (maybe adding in isobars for cyclone-centric orientation). In its current format, it shows less than what a 2D plot shows – but has clear potential to be excellent if it were better focussed*

*(I can see why you would want to see all of this in 3D, it just does not show up well). Figure A3 for cyclone Egon is actually clearer; however, A3 would also benefit from having a slightly more upright angle, removing the land and adding isobars.*

Response:
Again, thank you for your helpful suggestions. We agree that this figure is difficult to read and that it would benefit from splitting into 6 panels and slightly change the viewing angle.

Action:
- We split Figure 10 (and Figure A3) into two main panels (Fig. 12 in the revised manuscript):
  - To keep the information about the geographical location of the cyclone, the revised top panel shows the spatio-temporal evolution of the 3-D frontal surfaces, similar the old figure (slightly modified colours and view angle).
  - The second main panel shows 6 separated, cyclonic centered plots where the base map and graticule is replaced with surface pressure isobars (we think these plots convey the temporal development of the frontal structures in a much clearer way than the original depiction).

*RC2: Lines 554-568 and Figures 11, A4 and A5: I found this passage very difficult to read and follow. The 3D plots make things difficult to reconcile. I would suggest circling exactly where you want the reader to look instead of trying to describe it (lines 554-556). You could then say something simpler like, "There is a gap in the frontal surface between 700-600 hPa in the ECMWF data whereas the frontal surface is present in the COSMO simulation (circled in Figs 12a, b)." It just focuses the reader on the point you want them to look at. I also think it might be worth including Figure A5 in Figure 12 and even plotting the difference between the THETA_W fields between COSMO and ECMWF (and possibly likewise for THETA). The reason for that it that I'm not convinced by your "convection drives differences in the temperature gradient" argument. It is possible that the opposite is true e.g. the temperature gradient around 700 hPa is stronger in COSMO (i.e. simulated better), which then leads to the development of convection along the frontal zone. The front may have been going through frontogenesis and the convection is just the result of that. I therefore do not believe your description of this process is convincing enough to be certain of the process you describe. The analysis does not contain enough detail.*

Response:
Thanks for your comment. We agree that the described relationship between convection and frontal structure is hypothetical and currently not supported by sufficient results. In the original manuscript, we attempted to frame this as a hypothesis that needs further investigation. However, to avoid confusion, we have reworded this paragraph to avoid emphasizing the potential influence of convection on frontal structure.
Regardless of the detailed relationship between the occurrence of convection and detailed frontal structure, we believe that more systematic and detailed analyses are needed, which are now possible with the 3-D front detection available in Met3D.

Action:
- We adapted the text according to our response.

- Following your recommendations, we have slightly modified the figure and the according paragraph:
    - Added the panels of Figure A5 into Figure 12 and added a circle in Figure 12.
    - Removed figure A5 from the appendix since it is obsoleted.
    - In Figure 12, a circle has been added to focus the reader on the area between 700 and 600 hPa.
    - We have reworded this paragraph to avoid emphasizing the potential influence of convection on frontal structure.

**Minor points:**

*RC2: Figure A1: I cannot find any reference to this figure in the text. Can it be removed?*
Response: We agree, Fig. A1 has been removed from the appendix.

*RC2: Figure A2: I think this can be removed if you adjust Figure 9 (you could even include the jet in figure 9).*
Response: Thanks for this suggestion, the jet stream is now included in Figure 9f. Figure A2 has been removed from the appendix.

*RC2: L221-224: Sentence starting "The method uses..." is very long. Please split this in two.*
Response: Thank you, the sentence is spitted into two.

*RC2: L287: Change "a decay stages" to "a decay stage".*
Response: Thank you, corrected.

*RC2: L300: Change "UTC5.3" to "UTC".*
Response: Thank you, removed "5.3".

*RC2: Figure 3 caption: remove the extra ")5.3" near the end of the third line.*
Response: Thank you, removed ")5.3".

*RC2: Line 321: Add "on" before "17 January 2018".*
Response: Thank you, added "on".

*RC2: Line 324: Change "As a result of the cyclone, high wind speeds were registered..." to " The cyclone caused high wind speeds..." to be more concise.*
Response: Thank you. Changed the sentence as suggested.

*RC2: Line 328: Change "... this is a Shapiro... " to "... this was a Shapiro..." as it happened in the past.*
Response: Thank you, changed the "is" to "was".

*RC2: Figure 7: Please include the time and date for these plots. It helps for stopping the video in the relevant place (I can see this information is in Figure 2 but should also be here). Also, the caption is unnecessarily detailed as you say most of it in the text. The caption only needs*

*the description of the figures, not the explanation about what each step does (as you explain in the text). Please trim the caption down to make it easier to read.*
Response: Thank you for your feedback. We added the date and time and the figure caption is shortened.

*RC2: Figure 7c: Maybe I'm missing something, but it looks like the feature is still in the plot under the blue circle (unlike in 7e where the northern feature disappears). Is this figure correct? Additional – Line 429 (related to Fig 7c comment) – OK I see this more now, but it is very subtle. I would focus that blue circle in to EXACTLY where you want the reader to look.*
Response: Thank you for your feedback. We agree, the blue circle in Fig. 7c was too large. In the revised manuscript, the blue circle focusses on a smaller area and a vertical pole was added in this area to help the reader notice the changes of the frontal surface.

*RC2: Figures 7d: it is very hard to work out where this cross section is taken without looking at 6f as there's too much shading. If you put a line on Figure 6c to show the location of the cross-section then that would help make it clearer (then refer to it in the caption).*
Response: Thank you for your feedback. We added the cross-section in Figure 7c and 7e, slightly changed the angle of the viewpoint in Figure 7d and 7f and slightly adapted the data area.

*RC2: Line 427: should it be "of the filter" instead of "of filter"?*
Response: Thank you, removed "the".

*RC2: Figure 9: The plots get very 'busy' with time, especially figure (f). If you could show where the viewpoint 2 and viewpoint 3 cross sections are located in figure (a) for example, then it'd help. If the land masses weren't blocked out so much in / at the very periphery of figure (f) then that would help.*
Response: We have changed the old Figure 9 according to your suggestions under "major points" above. The new figure (Fig. 8 in the revised manuscript) only uses a single view point.

*RC2: Line 475: "north-easterly direction" should really be "north-eastward direction".*
Response: Thank you, corrected.

*RC2: Fig. 11: For clarity, it might be worth making the THETA_W scale blue-red so that the grey shading shows up better. The grey-blue end could be confused with the horizontal gradient shading.*
Response: Thank you for your comment. We slightly adapted and reduced the smoothness of the gradient colour bar. This hopefully helps to distinguish between gradient shading and theta_w.

*RC2: L585: Change "The most easterly front, ranging from..." to "The most eastward front, extending from...".*
Response: Thank you, corrected.

*RC2: L587: Do you mean blue tubes for Fig 13b not green?*
Response: Yes, thank you, changed "green" to "blue".

*RC2: Lines 594-595: I do not see the need to describe the feature between 700 hPa and 500 hPa as it has no relevance to what you are focusing on (i.e. the low-level THETA_W feature). Please remove.*

Response: Thank you, we removed this sentence from the manuscript.

*RC2: L647-648: "We find that cold frontal...", I disagree with this sentence because you have not shown this. The description of the case study is not detailed enough to be certain of this reasoning (as mentioned in the major points).*

Response: Thank you for your comment. We agree that this case study is not sufficient for this statement. Since we decided to keep the scope of the article as it is (see above), we have labeled this statement as a hypothetical assumption.